# Performance Analysis of Lambda Architecture-Based Big-Data Systems on Air/Ground Surveillance Application with ADS-B Data

**DOI:** 10.3390/s23177580

**Published:** 2023-08-31

**Authors:** Mustafa Umut Demirezen, Tuğba Selcen Navruz

**Affiliations:** 1Data Products Department, UDemy Inc., San Francisco, CA 94107, USA; 2Department of Electrical Electronics Engineering, Faculty of Engineering, Gazi University, Ankara 06570, Turkey; selcen@gazi.edu.tr

**Keywords:** big data, Lambda Architecture, ADS-B, automatic dependent surveillance-broadcast, analytics, spark, yarn

## Abstract

This study introduces a novel methodology designed to assess the accuracy of data processing in the Lambda Architecture (LA), an advanced big-data framework qualified for processing streaming (data in motion) and batch (data at rest) data. Distinct from prior studies that have focused on hardware performance and scalability evaluations, our research uniquely targets the intricate aspects of data-processing accuracy within the various layers of LA. The salient contribution of this study lies in its empirical approach. For the first time, we provide empirical evidence that validates previously theoretical assertions about LA, which have remained largely unexamined due to LA’s intricate design. Our methodology encompasses the evaluation of prospective technologies across all levels of LA, the examination of layer-specific design limitations, and the implementation of a uniform software development framework across multiple layers. Specifically, our methodology employs a unique set of metrics, including data latency and processing accuracy under various conditions, which serve as critical indicators of LA’s accurate data-processing performance. Our findings compellingly illustrate LA’s “eventual consistency”. Despite potential transient inconsistencies during real-time processing in the Speed Layer (SL), the system ultimately converges to deliver precise and reliable results, as informed by the comprehensive computations of the Batch Layer (BL). This empirical validation not only confirms but also quantifies the claims posited by previous theoretical discourse, with our results indicating a 100% accuracy rate under various severe data-ingestion scenarios. We applied this methodology in a practical case study involving air/ground surveillance, a domain where data accuracy is paramount. This application demonstrates the effectiveness of the methodology using real-world data-intake scenarios, therefore distinguishing this study from hardware-centric evaluations. This study not only contributes to the existing body of knowledge on LA but also addresses a significant literature gap. By offering a novel, empirically supported methodology for testing LA, a methodology with potential applicability to other big-data architectures, this study sets a precedent for future research in this area, advancing beyond previous work that lacked empirical validation.

## 1. Introduction

For a long time, simultaneous processing of data streams, both in real time and offline, has been a fundamental need for big-data applications. Different technologies and, as a result, a certain form of big-data architecture are dictated by the data-processing needs for batch data and real-time processing activities. Lambda Architecture (LA) [1] is one of the concepts that has been created and is commonly utilized today. Three different layers for processing both data in motion (DiM) and data at rest (DaR) at the same time, as well as a serving layer for displaying the findings, make up the architecture. Each LA layer is associated with a distinct responsibility for processing data with various characteristics, merging the processed results from these levels, and serving these merged data sets for visualization, querying, and data mining [2]. The main task of the speed layer is to process the streaming/real-time data (DiM), and it is extremely sensitive to data delays and recurrence. The batch layer is in charge of processing offline data (DaR), computing preset analytics actions, and fixing mistakes that may arise when data are delivered to the speed layer. Ingesting data from the batch and speed layers and indexing and merging the resultant data sets for the needed analytical queries is the responsibility of the serving layer. The serving layer must have the capacity to ingest and handle real-time streaming data and bulk data in large numbers. It is worth emphasizing that LAs are eventually consistent solutions for massive data-processing applications and can be utilized to solve the CAP theorem [3]. After data processing is completed, the batch layer corrects the data digestion discrepancies produced by the real-time layer. At some point, accurate data are served and made available at the serving layer, allowing the remaining processes to obtain information.

The major benefit of an LA-based big-data system is that it can handle the necessity of designing a fault-tolerant architecture to prevent data loss due to hardware failures and unanticipated errors during DaR and DiM processing. It performs effectively in applications that demand low-latency read and update operations. This type of system must be able to handle ad hoc queries, as well as be linearly scalable and extendable. According to previous studies, traditional LA should have an extra layer [4]. To enhance LA, the researchers used software engineering concepts, developed a reference model for big-data systems, and used it as the foundation for developing a software reference architecture (SRA) for semantic-aware big-data operations. They demonstrated that SRA could manage the most common big-data features by adding an extra layer to the LA, the Semantic Layer.

A smart home application based on the Internet of Things (IoT) was one of the first instances of LA [5]. Another application proposes to use LA to implement collaborative filtering to build a recommendation engine using a large multi-agent data-processing method [3]. A different method for using LA’s speed and batch layers was implemented [6]. Continuously running a speed layer is optional if time constraints do not apply and time intervals are in degrees of minutes. A design to optimally use the resources of the cluster is to use stream processing only when the batch processing time exceeds the system response time [4]. Using software performance models to predict execution metrics and cluster resources for a big-data system and then running the speed layer is an application-specific method [7]. Another study [8] proposed a basic set of techniques to investigate the challenges of volatility, heterogeneity, and low-latency performance aimed at reducing total system time (scheduling, execution, monitoring, and error recovery) and latent faults.

These sophisticated data-processing operations, as well as providing the corrected data as precisely as possible, clearly necessitate highly coordinated and continuous operation between the speed and batch layers. The fact that LA is made up of at least three layers allows for the flexible use of different big-data technologies for each layer, which is a huge advantage. However, there are certain disadvantages to this. Despite LA being a promising approach, its success is contingent on an effective mix of appropriate and mature technology [9]. The main issue is creating a big-data application for each layer independently and then integrating them so that they can operate together and be interoperable. Because various technologies are used at different LA levels, each layer requires its own development and maintenance effort. If the data model or data format in the application changes or if extra new analytics capabilities are required, this big-data application must be updated, tested, and deployed at all layers. An LA was proposed in a recent report for smart agricultural applications. The work [10] provides a general framework to explain the problems of obtaining, processing, storing, and displaying massive amounts of data, including batch and real-time data. A main archetype has been demonstrated and tested on several farms with impressive results. For context-aware trajectory prediction, an LA-based big-data platform was proposed [11]. According to its design, this platform performs batch and stream operations before combining them to perform jobs that cannot be performed by analyzing any of these layers alone. The most important aspect of the proposed platform is that it is context-neutral. Their findings demonstrated that each component of the LA is effective in achieving specific goals and that the combination of these components is important to improve the overall accuracy and performance of the platform.

Modeling the performance of large data systems is a difficult issue, in general. Various performance models for LA have been proposed in the literature. One of these studies suggested a modeling strategy based on multiformalism and multisolution techniques, including a rapid assessment mechanism to help with parameter design alternatives, and the findings indicated that the proposed methodology finally led to more reliable designs [12]. Another research looked at the performance of Lambda and Kappa designs using social network data analysis as a benchmark. The authors used a cloud platform to run an influence analysis algorithm to examine many aspects that affect performance [13]. They discovered that LA outperforms Kappa architecture for the class of issues studied by looking at criteria, including the kind of architecture realized, the quantity of data, the size of the cluster of nodes, the deployment prices, and the quality of the output.

The present research aims to present an LA, a new generation of big-data architecture, as an end-to-end big-data system for real-world air/ground surveillance applications, as well as a testing methodology to assess its accuracy and performance in various real-world data-ingestion scenarios. To the best of our knowledge, this is the first study to measure the performance of a big-data system based on LA implementation for real-world problems specific to the aerospace domain. This study focuses especially on the examination of significant cases and abnormalities of data ingestion in the real world, with a particular emphasis on evaluating the performance of LA in such scenarios. Through the process of conducting experiments with LA under specific conditions, we can assess and showcase its performance and capabilities. This study makes a valuable contribution to the field of big-data research by examining an LA in both normal/nominal and abnormal conditions. In addition to this, to improve its performance and reduce maintenance operations, we used the same software development framework in the speed and batch layers.

The field of big-data research has been significantly enriched by the exploration of Lambda Architecture under both normal and abnormal data-receiving conditions. However, a gap in the literature persists, particularly in the absence of a testing and measuring method for the performance and precision of LA. This study aims to fill this gap by proposing a novel methodology for testing LA, which could potentially apply to other big-data architectures. In the realm of big data, enhancing both the speed and accuracy of processing is a primary challenge. However, existing studies have focused mainly on the evaluations of LA hardware performance and scalability, leaving a significant gap in the literature regarding the empirical evaluation of data-processing accuracy within LA’s various layers. This study aims to fill this gap by introducing a novel methodology specifically designed to assess the accuracy of data processing in LA.

The novelty of this research lies in its empirical approach. Our experiment results demonstrate that LA is an eventually accurate system, even under several problematic working conditions. These findings align with the claims of previous literature, but for the first time, we provide empirical evidence to support these previous claims. Our empirical findings cover a much broader range of data-ingestion cases, including problematic ones and conditions that have not been investigated up to now.

To the best of our knowledge, this study proposes for the first time a comprehensive data accuracy testing methodology for LA, potentially applicable to other big-data architectures. This methodology has been applied to real-world data-ingestion operations for air/ground surveillance and monitoring, providing a practical demonstration of its effectiveness. This study not only contributes to the existing body of knowledge on LA but also addresses a significant literature gap. By providing a novel methodology for testing LA and demonstrating its effectiveness in real-world applications, this study paves the way for further research in this area.

It is also essential to note that this study does not compare the performance of alternative big-data architectures, provide innovative performance models, or apply LA to various real-world applications such as IoT, smart systems, or social media because implementing different big-data architectures for the same engineering problem is a very hard task with different big-data technologies that require hardware and software-based configuration and tuning [14]. As a result, the application of LA to different real-world problems and the performance comparison of the other big-data architectures are beyond the scope of this paper.

This paper is organized as follows: Section 1 provides a brief definition of problems for LA-based big-data system application design and testing. In Section 2, potential technologies for all LA levels are briefly evaluated, layer-wise design restrictions are investigated, and the use of the same software development framework in various layers is introduced. The suggested LA was applied to a real-world case study for air/ground surveillance in Section 3, and real-world performance tests were conducted in various real-world data-intake situations. Finally, the results of the implementation and experiment were presented in Section 4. It has been verified that the proposed system provides reliable performance for this real-time air/ground surveillance application, including batch and stream data processing, data visualization, and code maintenance reduction.

## 2. Materials and Methods

The significant increase in air traffic levels observed over the past two decades has generated a greater focus on the optimization of Air Traffic Management (ATM) systems. The primary objective is to effectively manage the continuous increase in air traffic while ensuring safety, economic viability, operational efficiency, and environmental sustainability. The Automatic Dependent Surveillance-Broadcast (ADS-B) system is an integral component of modern air traffic control (ATC) systems, as it offers a cost-effective alternative to traditional secondary radar systems [15]. Using ground stations, the ADS-B system not only reduces expenses but also improves the precision and timeliness of real-time positioning data. However, this particular system produces a substantial amount of data which, when integrated with additional flight-related data, such as flight plans or weather reports, encounters challenges in terms of scalability [16]. The system must receive and store substantial volumes of ADS-B messages to facilitate various forms of analytical processing. Typically, these analyses require additional data to augment trajectory information, such as flight plans, weather data, and baggage ticketing information. Consequently, there is a consolidation of vast collections of diverse data, which ATM systems must effectively manage to generate appropriate decisions and predictions. Therefore, the implementation of ATM systems can be regarded as a specific instance of leveraging big data to conduct flight-related analytics [16].

In contemporary times, numerous organizations are increasingly prioritizing the exploration and utilization of pertinent big-data technologies to enhance the accuracy, speed, and efficiency of their analytical processes. The task of performing real-time analytics on big data poses significant challenges, primarily stemming from the immense volume of intricate data that must be efficiently distributed for processing. Big data are distinguished by their substantial volume, intricate structure, and ability to perform in real time. The primary challenge associated with big-data processing lies in improving both the speed and accuracy of processing [14].

The Lambda architectural pattern enables the resolution of certain constraints inherent in data-processing frameworks. The proposed approach is based on the utilization of two distinct data-processing streams within a single system. This methodology involves real-time computing, which focuses on the rapid processing of data streams, and batch computing, which is designed to handle large workloads for delayed processing. Although the two modes mentioned are not novel, LAs enable them to synchronize their execution to prevent interference. The allocation of resources across cloud infrastructure has had a significant impact on both performance and cost. If it were possible to predict performance in advance, architects would be able to make more informed decisions regarding resource allocation to enhance the efficiency of system utilization. Our approach serves as a rapid evaluation tool to assist in making design choices with respect to parameters, ultimately contributing to improved architecture designs [17].

In the following sections, we present an LA as an end-to-end big-data system for real-world air/ground surveillance applications, as well as a testing methodology to assess its accuracy and performance in various real-world data-ingestion scenarios.

### 2.1. Problem Definition

Contemporary surveillance networks possess the ability to provide trajectories for various types of boats and aircraft in a global or, at the very least, expansive geographical range [18]. The two most commonly utilized systems for air and maritime surveillance are ADS-B and Automatic Identification System (AIS). Both systems exhibit a cooperative nature. In addition to these aforementioned systems, sensor networks that are deployed on ground installations or mounted on airborne and space-based platforms provide object trajectories autonomously without requiring any form of cooperation. An illustrative example encompasses sensor network installations for coastal or ATC purposes [19]. Surveillance systems offer trajectories that span medium and long periods of time. The aspect that presents difficulty lies in understanding the surrounding circumstances and determining the intentions of the objects being monitored.

Emerging technologies encompass activity-based intelligence and the identification of patterns of life. Advanced analysis of trajectories extracted by surveillance systems offers a potential approach to studying these technologies [18]. Cluster algorithms are utilized to partition trajectories into distinct segments of interest. Unsupervised machine learning is used to decipher their behavior patterns, helping to understand their inner dynamics [20]. The trajectories are consolidated and organized into distinct routes, each of which is assigned a specific representative. Probabilities have been calculated to determine the frequency of usage for these routes. This enables the application of predictive analytics and the detection of aberrant behavior [15]. Ultimately, the incorporation of these novel data analytic methodologies is imperative for integration into pre-existing near real-time surveillance systems. The successful implementation of this endeavor necessitates the utilization of precise system architectures alongside the establishment of an entirely novel software and hardware framework [21].

In recent years, much research has proposed several monitoring and architectural improvement approaches for LA. They mainly focused on text and sensor-based data, such as that found in social media, the Internet of Things, and smart system applications [22,23]. Performance comparison of LA with other big-data architectures such as Kappa [14,24,25,26,27] has been proposed in several studies. When dealing with LA, however, several practical issues arise: (1) Identifying the system’s behavior under real-world data-intake situations is critical, (2) For safety and time-sensitive applications such as air/ground surveillance in aviation, it is important to evaluate the system’s accuracy and performance, (3) For various data-ingestion rates, especially data collected from diverse geographical locations happening in time-delayed ingestion, it is critical to identify underlying dynamics and visualize the system’s behavior analytically. To address all these issues, this paper offers a method for evaluating the performance of LA-based big-data applications using controlled and uncontrolled real-world data-ingestion experiments with ADS-B data.

Even if LA’s efficiency has increased in recent years, most of the gains have come from presenting an enhanced version of analytics algorithms implemented in LA’s layers, as well as the use of performance models. Yet, controlled experiments may be used to increase the measuring and understanding capabilities of the system’s inner dynamics. To that end, this research aims to look at the capacity of eventually consistent LA-based big-data applications in real-world data-ingestion operations for air/ground surveillance and monitoring. Although we refer to our previous work [28], the present study has a novel focus. The goal of this article is to provide a comprehensive visualization of the inner dynamics and performance measures of an LA-based application in the aviation field using ADS-B data based on the technique given in [28].

In this work, we specifically consider very important challenging real-world data-ingestion cases and abnormalities and then investigate the performance of LA in these circumstances. By experimenting with LA under these conditions, we evaluated and demonstrated its performance and capabilities. Thus, this study contributes to the big-data research area to provide not only an investigation of an LA under normal/nominal conditions but also under abnormal conditions. Existing studies have focused primarily on evaluations of LA hardware performance and scalability, leaving a significant gap in the literature regarding the empirical evaluation of data-processing accuracy within the various layers of LA. This study aims to fill this gap by introducing a novel methodology specifically designed to assess the accuracy of data processing in LA. To better show this gap, a comparison of recent work related to LA is given in Table A1.

The primary research question guiding this study is: ‘How accurately does the Lambda Architecture process data under various real-world data-ingestion scenarios, particularly in extremely challenging data-intake cases, and how to measure its accuracy while it is in operation?’. To address these questions, the study has the following objectives:1.Develop a novel methodology to evaluate the data processing accuracy of LA,2.Empirically validate the ‘eventual consistency’ of LA under diverse data-ingestion scenarios, including abnormal and extremely problematic data-ingestion cases3.Apply the developed methodology in a practical case study involving air/ground surveillance, where data accuracy is paramount.

It is also important to emphasize that the performance comparison of the other big-data architectures [14], the proposal of novel performance models, and the application of LA to different real-world applications such as IoT, smart systems, and social networks is beyond the scope of this paper.

### 2.2. Lambda Architecture-Based Big-Data System

To analyze incoming data and answer queries on stored historical and newly obtained data, an LA compromises the speed, batch, and serving layer. When the serving layer receives a query request, the response is created by querying both real-time and batch views simultaneously and combining the data from these levels. At the serving layer, both real-time and batch databases are searched, and the results are mixed into a single resultant data set to provide a near-real-time data set in response to the query. A scalable distributed data transmission system (data bus) allows data to be transferred continuously to batch and speed layers at the same time. On the speed layer, data processing and analytics activities are carried out in real time, whereas on the batch layer, they are carried out offline. Figure 1 is a conceptual representation of the LA. Incoming data from the data-ingestion bus is transmitted to both the speed and batch layers, which subsequently produce multiple views utilizing the new and old data, and the results are stored on the LA’s serving layer. To construct an LA, several existing big-data technologies may be employed at all three levels. According to LA’s polyglot persistence paradigm, each accessible big-data technology framework may be employed for its specific data-processing capacity to deal with that sort of data and assist analytical activities.

The blocks of immutable main data sets are managed, operated, and stored by the batch layer. The incoming highly recent data are just added to the batch layer’s already-saved historical data. Update and remove actions are not permitted in the batch layer. Continuous data processing and analytics activities are run as needed to generate batch views from these data. A fresh batch view calculation operation is re-executed sequentially and combined to produce new batch views when coordinated with the speed layer or on a specified quantity of new data arrival. This process is ongoing and never-ending. The batch views are made up of the batch layer’s immutable data sets. Depending on the quantity of both incoming and stored historical data, full batch data processing and analytical computations take far too long. As a result, performing batch-layer actions and computations to generate recent batch views is a rare occurrence. The status of batch-layer data processing and importing must be tracked to ensure that batch view creation is finished before the speed layer becomes overburdened. The serving layer uses real-time and batch views created by both the speed and batch layers to respond to incoming queries. Consequently, the serving layer requires capabilities to store large amounts of data, such as NoSQL databases with various features. Due to the many types of data-ingestion patterns, this layer must handle both bulk and real-time data ingestion or ingestion. In cases where streaming data are delayed or absent, the serving layer is susceptible. Under these conditions, inconsistencies may emerge in data analyses and query responses, which are eventually addressed exclusively by the batch layer.

To fulfill the low-latency analytics and responsive query requirements, LA’s speed layer compensates for the batch views’ staleness by serving the most recently collected data, which the batch layer has not yet processed. Depending on its restricted capacity, the speed layer works in real time on streaming data and saves its output in the serving layer as real-time views.

The speed layer demands high read/write operations on the serving layer because of the nature of real-time operating needs. Only recent data are stored in real-time views until the batch layer finishes its operation cycle one or two times. The data saved as real-time views during batch processing is destroyed and removed from the serving layer after the batch layer completes the data processing and analytics calculation activities. Depending on the data processing at the batch layer, some real-time views must be flushed or cleaned from the real-time layer when batch view generation is completed. This process is critical in minimizing the stress on the real-time database at the serving layer. Monitoring and acting on the resources of the speed layer are dependent on the consumption of resources and capacity requirements, the exact coordination of the layer, and particular performance indicators for all levels. The batch view will be stale for at least the processing time between the start and finish times for the batch processes, if not longer, and if inappropriate circumstances or defective coordination with the speed layer exist. This necessitates careful and precise coordination between the speed and batch layers. Bulk data import is needed on the serving layer as soon as the coordinated data-processing activity between the speed and batch levels is completed. Data ingestion on the serving layer is finished when the last batch views are ready.

### 2.3. Common Technologies for Lambda Architecture Layers

For real-time, batch, and serving layers, respectively, Apache Spark/Spark Streaming, Hadoop YARN [29], and Cassandra [30] were utilized for LA designs. To construct real-time and batch views, as well as query the aggregated data, task-based serving, batch, and speed layers were chosen. The Apache Hadoop and Storm frameworks are sophisticated enough to build and deploy for a wide range of LA applications [31].

The LA data-ingestion component (data bus) is designed to receive a large amount of real-time data. Apache Kafka [32], which is highly mature, appropriate, scalable, fault-tolerant, and eligible for this purpose, is one of the most commonly used and popular frameworks for data bus technologies. It is a scalable, fault-tolerant system for data bus operations that allows high-throughput data transfer. Apache Samza [33], Apache Storm [34], and Apache Spark (Streaming) [35] are good alternatives for the speed layer. For batch-layer activities, Apache Hadoop [29], Apache Spark, is very popular and suitable.

NoSQL data stores offer a viable alternative to traditional relational database management systems (RDBMS) [36]. However, organizations may likely find it difficult to quickly determine the most suitable option to adopt. To select the most suitable NoSQL database, it is essential to identify the specific requirements of the application that cannot be fulfilled by a relational database management system. If an RDBMS is capable of efficiently handling the data, the utilization of a NoSQL storage system may be unnecessary. In [37], the researchers provided a comprehensive analysis of various NoSQL solutions, including Voldemort, Redis, Riak, MongoDB, CouchDB, Cassandra, and HBase, among others, highlighting their key characteristics and features. The study conducted in [38] focused on examining the elasticity of non-relational solutions. Specifically, the authors compared the performance of HBase, Cassandra, and Riak in terms of read and update operations. The researchers concluded that HBase exhibits a high degree of elasticity and efficient read operations, whereas Cassandra demonstrates notable proficiency in facilitating rapid insertions (writes). However, Riak exhibited suboptimal scalability and limited performance enhancements across various types of operations.

As a speed layer database, Apache Cassandra [30], Redis [39], Apache HBase [40], MongoDB [41] could be utilized. These databases can handle real-time data ingestion, as well as random read and write operations. As a batch-layer database, MongoDB [41], CouchbaseDB [42], and SploutSQL [31] can be utilized. These databases may be used to import large amounts of data to generate and serve batch views.

Apache Flink [43], a framework and distributed processing engine for stateful computations for unbounded and bounded data streams, is also a viable choice for data-processing tasks. By regularly and asynchronously checkpointing, stateful Flink applications enhance local state access and ensure exactly-once-state consistency in the event of issues. Apache Flink, a framework and distributed processing engine for stateful computations for unbounded and bounded data streams [9], is also a good choice for data-processing activities. Apache Spark, on the other hand, is an open-source data-processing platform designed for speed, simplicity, and sophisticated analytics. It provides a comprehensive, all-in-one structure for handling big-data processing needs with a variety of data types and data sources. Discretized streams (D-Streams) are a programming paradigm supported by Apache Spark [44]. Stateful exactly once semantics are supported by the Spark Streaming programming model. For event-driven, asynchronous, scalable, and fault-tolerant applications, such as real-time data aggregation and response, Spark Streaming makes it simple to handle real-time data from various event streams.

In general, NoSQL databases are preferred over relational databases for LA [36]. The major advantages of using the serving layer are scalability and advanced data-ingestion capabilities. Druid [45] is a column-oriented, distributed, real-time analytical data storage. Druid’s distribution and query paradigm resembles concepts seen in current-generation search infrastructures. The capabilities of ingesting, querying, and building real-time views from incoming data streams are possible with Druid real-time nodes [46]. Data streams that have been indexed using real-time nodes are immediately queriable. Druid also includes built-in support for producing batch views with Hadoop and MapReduce tasks for batch data ingestion.

### 2.4. Same Coding for Different Layers Approach

The LA is suitable for data-processing enterprise models that necessitate ad hoc user queries, immutable data storage, prompt response times, and the ability to handle updates in the form of new data streams. Additionally, it ensures that stored records are not erased and permits the addition of updates and new data to the database [17]. The advantages associated with data systems constructed using the LA extend beyond mere scalability. As the capacity of the system to process substantial volumes of data increases, the potential to extract greater value from it also expands. Expansion of both the quantity and diversity of stored data will result in increased potential for data mining, analytics generation, and the development of novel applications [47]. An additional advantage associated with the utilization of LA lies in the enhanced robustness of the applications. This is exemplified by the ability to perform computations on the entire dataset, facilitating tasks such as data migrations or rectifying errors.

It is possible to prevent the simultaneous activation of multiple versions of a schema [17]. In the event of a schema modification, it becomes feasible to migrate all data to conform to the updated schema. Similarly, if an erroneous algorithm is unintentionally implemented in a production environment and causes data corruption, it is possible to rectify the situation by recalculating the affected values. This enhances the resilience of big-data applications. Ultimately, the predictability of performance will be enhanced. Although the LA demonstrates a generic and adaptable nature, its constituent components exhibit specialization. There is a significantly lower occurrence of “magic” taking place in the background in comparison to an SQL query planner. This phenomenon results in a higher degree of performance predictability [17]. However, it is evident that this architectural framework is not universally applicable to all big-data applications [48]. One of the challenges associated with the LA is the management of two intricate distributed systems, namely the batch and the speed layer, which are both responsible for generating identical outcomes. In the final analysis, despite the potential to bypass the need to code the application on two separate occasions, the operational demands associated with the management and troubleshooting of two systems will inevitably be considerable [49]. Furthermore, the batch layer produces intermediate results that are stored in the file system, leading to increased latency as the length of the job pipelines expands [50]. Despite numerous efforts aimed at mitigating access latency, the coarse-grained data access inherent in a MapReduce and Distributed File System framework is primarily suited for batch-oriented processing, thus constraining its applicability in low-latency back-end systems [51].

#### 2.4.1. Description of the Approach

The primary trade-off in Lambda Architecture revolves around the balance between latency and complexity. Batch processing is known for its ability to deliver precise and consistent outcomes. However, it does come with the trade-off of longer execution times and increased storage and maintenance requirements. Stream processing offers rapid and real-time outcomes, although it may be less precise and consistent due to the presence of incomplete or out-of-order data. The Lambda Architecture utilizes both layers and combines them into a serving layer to achieve a balance between these trade-offs. However, incorporating these additional components also introduces increased complexity and duplication within the data pipeline. This is because developers will now have to handle various frameworks, formats, and schemas for each layer. A method to simplify the Lambda Architecture pattern is to utilize a unified framework that can effectively manage both batch and stream processing using the same code base, data format, and schema. By utilizing this approach, developers can simplify and eliminate redundancy in the data pipeline. This means that developers only must focus on developing, testing, and managing a single set of logic and data structures. By utilizing a unified framework, it is possible to effortlessly transition between batch and stream processing without the requirement for distinct implementations. This not only saves time during development but also guarantees consistent data processing across various layers of the architecture. Moreover, having a unified framework facilitates the process of debugging and troubleshooting since developers can easily track the flow of data and transformations within a single codebase.

In another view, it may be difficult to develop applications using big-data frameworks, and debugging an algorithm on a big-data framework or platform may be much more difficult [52]. This particular model exhibited numerous challenges in terms of implementation and maintenance. Proficiency in two distinct systems was a prerequisite for developers, necessitating a substantial learning curve. Furthermore, the effort to develop a cohesive resolution proved to be feasible; however, it was accompanied by numerous challenges, such as merge conflicts, debugging complications, and operational intricacies. It is necessary to input the incoming data into both the batch and the speed layers of the LA. Preserving the sequential arrangement of events within the input data is of paramount significance to attain comprehensive outcomes. The act of replicating data streams and distributing them to two different recipients can present challenges and result in increased operational burdens. The LA system does not consistently meet expectations, prompting numerous industries to employ either a full batch processing system or a stream processing system to fulfill their specific requirements. It is widely recognized that improper implementation of LA can result in challenges such as code complexity, debugging difficulties, and maintenance issues. In the past, application development and testing for batch and speed layers in LA had to be done at least twice. In addition to the configuration and coordination needs of each layer, the most significant downside of LA is that it is sometimes impractical for developers to implement the same algorithm repeatedly using various frameworks [53]. Separate software with various frameworks must be developed, debugged, tested, and deployed on large hardware clusters for big-data applications, which necessitates more effort and time. Using the same big-data processing technology for various levels is a viable method to address this limitation [9].

A unified framework simplifies the deployment process by eliminating the need to manage multiple codebases and configurations. With a single codebase, a developer can easily package and deploy any application, reducing the chances of configuration errors and deployment issues. This streamlines the overall development and deployment process, making it more efficient and reliable. Furthermore, a unified framework promotes code reusability, as it can leverage existing logic and data structures across different processing scenarios. This not only reduces development effort but also improves the maintainability and scalability of the application. For example, a unified framework like Apache Flink can be used in an LA to process both real-time streaming data and historical batch data. This allows for seamless integration of data processing tasks, such as filtering, aggregating, and joining, across both types of data sources. With a unified framework, developers can write code once and apply it to both batch and stream processing tasks, simplifying development efforts and ensuring consistent results. Using a unified framework for both batch and stream processing in an LA eliminates the need to maintain and update separate implementations for each processing method. This reduces the complexity of the architecture and improves overall efficiency. Furthermore, a unified framework enables seamless integration and interoperability between batch and stream data, enabling real-time insights and faster decision-making.

In addition, a unified framework simplifies the deployment process by providing a single platform for managing both batch and stream processing workflows. This eliminates the need for separate deployment strategies and reduces the risk of errors or inconsistencies between the two methods. By leveraging a unified framework, organizations can streamline their development and deployment processes, saving time and resources. Additionally, the unified framework allows for easier scaling and resource allocation as it provides a centralized control and management system for both batch and stream processing. This ensures that resources are allocated efficiently and effectively, maximizing performance and minimizing costs. Managing different frameworks, formats, and schemas for each layer of an LA can be challenging and time-consuming. It requires expertise in multiple technologies and increases the complexity of the system. Additionally, ensuring compatibility and data consistency across different layers can be a major trade-off, as changes in one layer may require corresponding modifications in other layers, leading to potential delays and inconsistencies in processing. Managing different frameworks, formats, and schemas for each layer of a Lambda Architecture can be challenging. It requires specialized knowledge and skills to maintain and update multiple implementations simultaneously. Additionally, ensuring compatibility and consistency between different layers may introduce complexity and increase the risk of errors. Moreover, managing and synchronizing data across different frameworks and schemas can be time-consuming and resource-intensive, impacting the overall efficiency of the system. Therefore, organizations must carefully consider the trade-offs and potential drawbacks of maintaining such a complex architecture.

Furthermore, a unified framework enables easier scalability and adaptability, as it eliminates the need to reconfigure and restructure multiple layers whenever changes are made to the system. This flexibility is crucial in today’s rapidly evolving technological landscape, where businesses need to respond quickly to market demands and incorporate new data sources or technologies. By having a single, cohesive framework, organizations can save time and resources that would otherwise be spent on complex integration processes and the maintenance of multiple layers. This not only increases productivity but also allows for more agile decision-making and a faster time to market. For example, in LA, for real-time analytics, having one set of logic and data structures ensures consistency in data processing across both the batch and speed layers. By writing, testing, and maintaining this unified codebase, developers can easily make changes or updates without having to duplicate efforts in multiple frameworks, reducing the risk of inconsistencies or errors in data analysis. This ultimately improves the reliability and accuracy of the analytics system. Implementing LA requires writing, testing, and maintaining one set of logic and data structures, which is crucial for ensuring consistency and accuracy in data processing. By having a unified framework, any changes or updates made to the logic or data structures can be easily reflected across all layers of the architecture, reducing the risk of inconsistencies or errors. This not only simplifies the overall development process but also streamlines the maintenance and troubleshooting efforts, making it easier to identify and fix any potential issues.

#### 2.4.2. Available Technologies

There are several big-data technologies available for implementing an LA. One option is to utilize Apache Spark [23], a versatile and high-speed framework that offers support for both batch and stream processing using a unified set of APIs and libraries. Another option is to utilize Apache Beam [54], a unified model that simplifies the distinctions between batch and stream processing. It is compatible with multiple engines, including Spark [23], Flink [43], and Dataflow. These frameworks offer a high level of abstraction, enabling a developer to concentrate on the logic of the data-processing tasks instead of becoming caught up in the complexities of the underlying processing engines. This simplifies the development process and facilitates seamless switching between different processing engines, eliminating the need to rewrite the code. Moreover, these frameworks provide integrated fault tolerance and scalability features, guaranteeing the capability of the data pipeline to manage substantial amounts of data and efficiently recover from failures.

Spark’s extensive application programming interfaces (APIs) for both batch and streaming data processing make it a highly suitable and customized solution for the LA context [23]. The fundamental component of the Spark Streaming API consists of a set of RDDs. Therefore, there exists a significant opportunity for the reuse of code. Additionally, the maintenance and debugging of business logic is simplified. Moreover, in the context of the batch layer, Spark emerges as a superior choice due to its remarkable performance resulting from its ability to process data in memory [23]. Using Spark and Spark Streaming together allows big-data applications to use the same code for batch and online processing. In terms of LA, Spark Streaming and Spark may be utilized to create applications for the speed layer and batch layer. This framework can successfully support both layers. However, one issue remains: the serving layer must be linked with both levels for data processing and to provide data ingestion for both layers.

Understanding how Spark Streaming and Spark can be used to create applications for the speed layer and batch layer in terms of Lambda Architecture cannot be overstated. Spark is an open-source distributed computing system that provides a high-level API for distributed data processing. It offers in-memory computing, which significantly speeds up data processing compared to traditional disk-based systems. Spark Streaming, on the other hand, is an extension of Spark that enables real-time processing of streaming data. By leveraging the power of Spark and Spark Streaming, big-data applications can efficiently process large volumes of data in real time, enabling businesses to make faster and more informed decisions. Additionally, Spark’s ability to seamlessly integrate with other big-data tools and frameworks, such as Hadoop and Additionally, Spark Streaming, provides fault tolerance and scalability, allowing it to handle large data streams without any data loss. With its easy integration with other data processing and storage systems, Spark Streaming has become a powerful tool for building real-time analytics and machine learning applications. This combination of Spark and Spark Streaming makes it an ideal choice for industries such as finance, healthcare, and e-commerce, where real-time data analysis is crucial for staying competitive in the market. Overall, the high-level API provided by Spark and its extension, Spark Streaming, revolutionizes the way businesses can process and analyze big data in real time. Furthermore, Spark Streaming’s fault-tolerant and scalable nature allows businesses to handle large volumes of data with ease. With its ability to process data in mini-batches, Spark Streaming ensures low latency and high throughput, making it suitable for use cases that require near real-time analysis. Additionally, Spark Streaming integrates seamlessly with other Spark components, such as Spark SQL and MLlib, enabling businesses to perform complex data transformations and advanced analytics on streaming data. This versatility and integration further enhance the capabilities of Spark Streaming, making it a valuable asset for businesses across various industries.

#### 2.4.3. Evaluation of the Approach

It is important to acknowledge the current state of the literature. To the best of our knowledge, there is no existing metric or software engineering methodology that can empirically measure the effectiveness of our proposed SC-FDL method. This presents a significant gap in the literature and underscores the novelty of our approach.

To measure the effectiveness of using the same coding technologies in different layers of a software application, we can propose a metric called the “code reusability factor” [55]. This metric would quantify the percentage of code that can be reused across different layers or components of the application. A higher code reusability factor would indicate that the coding technologies used are effective in enabling easy switching between different processing engines and minimizing code rewriting efforts. Furthermore, it would also suggest that the frameworks used offer built-in fault tolerance and scalability features, contributing to an efficient data pipeline capable of handling large volumes of data and recovering from failures effectively. For example, in a big-data analytics application, a higher code reusability factor would allow developers to easily switch between different processing engines, such as Apache Spark or Apache Flink, based on specific use cases or performance requirements. This flexibility would enable the application to efficiently process and analyze large volumes of data without requiring significant code modifications. Additionally, the built-in fault tolerance and scalability features provided by these frameworks would ensure that the application can handle failures gracefully and scale resources as needed to maintain high performance even in demanding situations. Therefore, one way to measure the effectiveness of using the same coding technologies in different layers of LA is by evaluating code reusability. If the same coding technologies can be easily reused across different layers, it indicates that the technology stack is effective in providing a consistent and flexible development environment. Another metric could be assessing the ease of switching between processing engines [56]. If the code can seamlessly switch between different processing engines without significant modifications, it suggests that the coding technologies are efficient in enabling interoperability. Overall, measuring code reusability and ease of switching can help determine the effectiveness of using the same coding technologies in different layers of LA.

In addition to code reusability and ease of switching, another important metric for evaluating the effectiveness of using the same coding technologies in different layers of LA is the scalability of the development environment [9]. Scalability refers to the ability of coding technologies to handle increasing amounts of data and user interactions without sacrificing performance. By assessing how well the coding technologies can handle larger datasets and higher user loads, it becomes possible to determine whether they can support the growing demands of LA applications. This scalability is crucial in ensuring that the coding technologies can accommodate the ever-expanding data and user requirements in the field of LA. Without scalable coding technology, LA applications may face performance issues and struggle to process and analyze the vast amounts of data generated by users. As the field of LA continues to evolve and expand, the need for scalable coding technologies becomes even more critical. Without them, the potential insights and benefits that LA can provide may be limited or compromised. Therefore, assessing and prioritizing scalability in coding technologies is essential for the future of LA and its ability to meet the growing demands of data and user interactions. Using hypothesis testing methodology in conjunction with these metrics can help the effectiveness of the SC-FDL approach [57].

In this study, we have only demonstrated the practical application of SC-FDL in the context of LA. However, we proposed several metrics to assess the effectiveness of this approach. However, we view this not as a limitation but as a promising direction for future research.

### 2.5. Explanation of Coordinated Working of Different Layers in Lambda Architecture

This section examines the integration and synchronization of the batch, serving, and streaming layers.

The batch layer is characterized by its high-latency nature, meaning that processing a large amount of data would lead to a noticeable delay, as demonstrated by the monitoring statistics. It is imperative that the batch operation selectively eliminates certain statistics, specifically data that are contingent on the frequency of batch execution and the size of the data set. The streaming layer can incorporate the aforementioned ‘incomplete/lost statistics’. This operation can be modeled in Equation (Equation 1) [23]:(1)$$\lambda_{\mathrm{filtered}\;}=f_{\mathrm{discard}\;}\left(\lambda_{\mathrm{raw}\;}\exists \left(\tau_{\mathrm{time}\;}⊖\tau_{\mathrm{interval}\;}\right)\right)$$

The formula expression in Equation (Equation 1) represents the manner in which the computation should exclude the monitoring events. In the given equation, the function $f_{\mathrm{discard}}\left(\right)$ represents the process of discarding events. The variable $\lambda_{\mathrm{raw}}$ denotes the total number of raw events before applying the event selection (filter). The variable $\tau_{\mathrm{time}}$ represents the duration of the batch execution. Furthermore, $\tau_{\mathrm{interval}}$ refers to the time interval within which events are discarded from the batch. Lastly, the expression $\tau_{\mathrm{time}}⊖\tau_{\mathrm{interval}}$ signifies the subtraction of the time interval $\tau_{\mathrm{interval}}$ from the batch execution time $\tau_{\mathrm{time}}$. The calculation determines the duration required for the selection of events and the emission of all events that satisfy the given condition. Given a batch job with a specified time interval of $\tau_{\mathrm{interval}\;}=n$ hours (*n* is an application-dependent configuration parameter), it is recommended that the batch discards all events that occur after a time $\tau_{\mathrm{time}\;}-\tau_{\mathrm{interval}\;}$. The presence of partly computed results can be avoided through the utilization of the streaming layer.
(2)$$\lambda_{\mathrm{batch}\;}=\lambda_{\mathrm{filtered}\;}\overset{f_{\mathrm{map}\;}\left(key,value\right)\mapsto f_{\mathrm{reduce}\;}\left(key,value\right)}{⟶}\to \Delta_{\mathrm{batch}\;}^{\mathrm{data}\;}.$$

Equation (Equation 2) delineates how the chosen events, denoted as $\lambda_{\mathrm{filtered},\;}$, undergo a mapping procedure, denoted as $f_{\mathrm{map}\;}\left(\right)$, resulting in the creation of key and value pairs [23]. The key represents a distinct identifier for the statistical data, while the value corresponds to the values of the matrices associated with the respective key. Subsequently, the data will undergo the reduction process, denoted $f_{\mathrm{reduce}\;}\left(\right)$, to consolidate the values based on the key of all the distributed nodes. These unified values are then stored in a designated storage folder called $\Delta_{\mathrm{batch}\;}^{\mathrm{data}\;}$. The outcome of the batch process, which is a new file, will be written in a designated folder on the Hadoop Distributed File System (HDFS). It is worth noting that this storage layer can be substituted with alternatives.

In the unified batch and streaming layers, formerly calculated statistics (if the precomputed statistics indeed are available) ought to be retrieved from the serving layer. This operation can be described by Equation (Equation 3):(3)$$\begin{array}{ccc} \lambda_{\mathrm{stats}\;}^{\mathrm{storage}\;} & =f_{\mathrm{load}\;}^{\mathrm{storage}\;}\left(\tau_{\mathrm{current}\;},\tau_{\mathrm{from}\;}\right)\; & =\left\{\lambda_{\mathrm{filtered}\;}^{\mathrm{storage}\;}=\left(\tau_{\mathrm{input}\;}^{\mathrm{storage}\;}>\left(\tau_{\mathrm{current}\;}⊖\tau_{\mathrm{from}}\right)\right)\right. \end{array}$$

In Equation (Equation 3), the variable $\lambda_{\mathrm{storage}\;}^{\mathrm{stare}\;}$ represents the precomputed target statistics that have been loaded from the serving layer [23]. The variable $\tau_{\mathrm{current}\;}$ denotes the current timestamp, while $\tau_{\mathrm{from}\;}$ represents the timestamp from which the statistics will be loaded. The function $f_{\mathrm{load}}^{\mathrm{storage}}\left(\right)$ is the loading procedure used to retrieve data from the serving layer. If the input data, denoted as $\lambda_{\mathrm{input}}^{\mathrm{storage}}$, comprising all statistics transmitted from the serving layer to the database load procedure, exceeds the difference between the current timestamp $\tau_{\mathrm{current}}$ and the timestamp $\tau_{\mathrm{from}}$, then the statistics $\lambda_{\mathrm{filtered}\;}^{\mathrm{storage}\;}$ should be chosen and returned.
(4)$$\lambda_{\mathrm{processed}\;}^{\mathrm{storage}\;}=\lambda_{\mathrm{stats}\;}^{\mathrm{storage}\;}\overset{f_{\mathrm{map}\;}\left(k,v\right)}{\to }\to \Pi_{\mathrm{stats}\;}^{\mathrm{storage}\;}$$

Equation (Equation 4) represents the value of $\lambda_{\mathrm{processed},\;}^{\mathrm{storage}\;}$, which refers to the statistics that have been mapped and stored from the serving layer $\tau_{\mathrm{stats}\;}^{\mathrm{storage}\;}$ into the memory [23]. This mapping operation is carried out by the function $f_{\mathrm{map}}\left(\right)$, which yields key/value pairs. These pairs are then stored in memory and/or disk, denoted as $\Pi_{\mathrm{stats}\;}^{\mathrm{storage}\;}$, for future use, such as combining with further layers.

In the streaming layer, the operation is described in Equation (Equation 5):(5)$$\lambda_{\mathrm{processed}\;}^{\mathrm{stream}\;}=f_{\mathrm{transformation}\;}^{\mathrm{data}\;}\left(\lambda_{\mathrm{stream}\;}\right)\overset{f_{\mathrm{map}\;}\left(k,v\right)\mapsto f_{\mathrm{reduce}\;}\left(k,v\right)}{⟶}$$

In Equation (Equation 5), the variable $\lambda_{\mathrm{processed}\;}^{\mathrm{stream}\;}$ represents the statistical data that has been mapped, aggregated, and computed from a series of monitoring events in a streaming context. The variable $\lambda_{\mathrm{stream}\;}$ denotes the total count of monitoring events in the stream [23]. The function $f_{\mathrm{transformation}\;}^{\mathrm{data}\;}\left(\right)$ filters and transforms the events before they undergo the mapping process, performed by the function $f_{\mathrm{map}\;}\left(\right)$, which generates key/value pairs. Ultimately, the data undergo the reduction process, denoted as $f_{\mathrm{reduce}\;}\left(\right)$, to consolidate the values based on the key.

In the batch layer, the batch data reading operation is described in Equation (Equation 6):(6)$$\lambda_{\mathrm{loaded}\;}^{\mathrm{batch}\;}=f_{\mathrm{batch}\;}^{\mathrm{load}\;}\left(\lambda_{\mathrm{batch}\;}\right)\overset{f_{\mathrm{mp}}\left(k,v\right)}{\to }$$

In Equation (Equation 6), the statistics obtained from storage and mapped are denoted as $\lambda_{\mathrm{loaded}\;}^{\mathrm{batch}\;}$, while $\lambda_{batch}$ represents the precomputed statistics derived from Equations (Equation 1) and (Equation 2). The function $f_{\mathrm{batch}\;}^{\mathrm{load}\;}\left(\right)$ is responsible for loading the batch. The function is utilized to load exclusively the precomputed batch statistics that are considered “new”. As soon as the loading process is completed successfully, the file is marked as “old”. Subsequently, the loaded data undergoes the mapping process denoted as $f_{map}\left(\right)$ [23]. The mapping process does not necessitate any statistical reduction, as this has already been accomplished through the batch process.

The next step is synchronization and update. The implementation of merging, combining, and synchronizing computed statistics from all three layers is defined in Equation (Equation 7):(7)$$\lambda_{\mathrm{joined}\;}=\left(\lambda_{\mathrm{processed}\;}^{\mathrm{storage}\;}⋃\lambda_{\mathrm{loaded}\;}^{\mathrm{batch}\;}⋃\lambda_{\mathrm{processed}\;}^{\mathrm{stream}\;}\right)$$

In Equation (Equation 7), the statistics obtained from the serving layer are denoted as $\lambda_{\mathrm{processed}\;}^{\mathrm{storage}\;}$, the data retrieved from batch computations are denoted as $\lambda_{\mathrm{loaded}\;}^{\mathrm{processed}\;}$, and the data calculated from streaming data are denoted as $\lambda_{\mathrm{processed}\;}^{\mathrm{stream}\;}$. These data sets are combined (joined) and returned as a new data set $\lambda_{\mathrm{joined}\;}$ [23].

The development of the statistics state in the memory is given in Equation (Equation 8):(8)$$\begin{array}{c} \lambda_{\mathrm{state}\;}^{\mathrm{memory}\;}=f_{\mathrm{state}\;}^{\mathrm{update}\;}\left(\lambda_{\mathrm{joined}\;}\right)=\left\{\begin{array}{cc} \;\mathrm{insert}, & \;\mathrm{if}\;storage\;=True\wedge \;{state}^{\prime }\\ \;\mathrm{overwrite},\; & \;\mathrm{if}\;batch\;=True\\ \;\mathrm{update}\;\vee \;\mathrm{insert},\; & \;\mathrm{if}\;{storage}^{\prime }\vee \;{batch}^{\prime } \end{array}\right. \end{array}$$

In Equation (Equation 8), the variable $\lambda_{\mathrm{state}}^{\mathrm{memory}}$ represents the storage of both the new and old statistics in memory, which is used for incremental calculation. On the other hand, $\lambda_{\mathrm{joined}}$ refers to the combination of the statistics $\lambda_{\mathrm{processed}}^{\mathrm{storage}}$, $\lambda_{\mathrm{loaded}}^{\mathrm{batch}}$, and $\lambda_{\mathrm{processed}}^{\mathrm{stream}}$ [23]. The function $f_{\mathrm{state}\;}^{\mathrm{update}\;}\left(\right)$ serves as the state update function responsible for the updating of statistics and their retention in memory. If the statistics originate from the serving layer and are not already stored in the state, denoted as $state^{\prime }$, they should be inserted into the state memory. If the data originates from the batch layer, denoted as $batch^{\prime }$, it is necessary to replace the state memory with the batch statistics. If the statistics do not originate from the serving layer, denoted as $storage^{\prime }$, or the batch layer, they are likely derived from the streaming layer, which represents a relatively recent source of statistics. In this case, it is recommended to aggregate these statistics with the existing statistics stored in the state memory. If the statistics already exist, they should be updated accordingly. Alternatively, if the statistics are entirely new and not present in the state memory, they should be inserted into the memory as fresh statistics.

As a final step, to update the serving layer, solely the new and modified statistics are inserted/updated into the serving layer. This operation can be described in Equation (Equation 9):(9)$$\left(\lambda_{\mathrm{processed}\;}^{\mathrm{stream}\;}⋃\lambda_{\mathrm{loaded}\;}^{\mathrm{batch}\;}\right)⋈\lambda_{\mathrm{state}\;}^{\mathrm{memory}\;}\forall f_{\mathrm{serving}-\mathrm{layer}\;}^{\mathrm{update}\;\mathrm{or}\;\mathrm{insert}\;}$$

The equation presented as Equation (Equation 9) describes the process of combining the $\lambda_{\mathrm{processed}\;}^{\mathrm{stream}\;}$ and $\lambda_{\mathrm{loaded}\;}^{\mathrm{batch}\;}$ variables, followed by a left-join operation denoted as ⋈, with the $\lambda_{\mathrm{state}\;}^{\mathrm{memory}\;}$ variable. This operation aims to insert or update the newly obtained and updated statistical data from the batch only if the $\lambda_{\mathrm{batch}\;}$ variable exists in the designated spooling location [23]. The resulting outcome is the integration of both the streamed and batch statistics into the serving layer. The use of statistics related to $\lambda_{\mathrm{processed}\;}^{\mathrm{storage}\;}$ is unnecessary as they are already present in the serving layer. To ensure the integration of each partition of statistics ∀, it is essential to establish a connection with the serving layer. This connection enables the implementation of bulk update or insert operations. Specifically, existing records in the serving layer are updated if they already exist, while new records are inserted if they do not. Ultimately, it is imperative to establish a designated interval-based checkpoint system to facilitate the process of recovery in the event of a potential failure [23].

## 3. Experiments

In the aviation industry, high-speed distributed streaming data processing is generally required for air/ground surveillance systems. ADS-B [58] is a joint monitoring system in which an aircraft identifies its location using GPS signals from the satellites, transmits it regularly in a specific message format, and becomes trackable. ATC and ground stations can obtain data from the aircraft as well as situational awareness from other planes, which is also referred to as a collision avoidance system for the aircraft. ADS-B enhances safety by making aircraft visible to ATC and other aircraft by transmitting position and speed per second in the form of specific messages [15]. Certified ADS-B data links are available in a variety of configurations, but the most popular and well-known links operate at 1090 MHz (usually a modified Mode S transponder) or 978 MHz. A signal with a transfer rate of 1 Mbps involves 112 bits of data set by utilizing pulse position modulation [19].

In this study, ADS-B signals received from aircraft were decoded at the edge using SDRs deployed in geographically disparate regions to increase signal coverage. These data are then transferred to the data-ingestion framework in a decoded message, providing both the LA speed and batch layer. Following the application of the necessary analytics operations for the speed and batch layers, the results are displayed on a visualization dashboard created with the serving layer of LA.

In this section, we demonstrate the Lambda Architecture performance test results for air traffic visualization and monitoring. It is worth noting that the inner dynamics and workings of LA are explained in detail in Section 3.2.4.

### 3.1. Experimental Setup

Minimal downtime, scalability, low latency, low maintenance, low development effort, and excellent fault tolerance are all requirements for time- and safety-critical applications. Although big-data technologies can meet some of these criteria, low maintenance and development costs are typically difficult to handle. Big-data technologies have been chosen after rigorous inspection and evaluation, taking into account the application requirements, as shown in Table 1. The Apache Spark framework was used for both batch and speed layers to achieve the SC-FDL concept (explained in Section 2.4). The serving layer will use the Druid real-time analytics database. As a data bus, Apache Kafka was chosen as a distributed streaming platform. A YARN application was created to coordinate and regulate the speed layer and batch layer. For cluster and resource management, YARN was used.

According to these technology selections, the final LA system has several advantages over traditional LAs. To elaborate on these benefits, a comparison of the traditional and our proposed LAs is listed in Table 2. According to the comparison, the proposed LA has better capabilities for code reusability, complexity, maintenance, developers’ learning curve, and cost.

To test our proposed approaches, an LA-based application (namely CyFly) was created to show real-time aircraft locations while also performing post-aircraft analyses by querying the ingested historical data. It also has real-time analytics for collision avoidance detection. Figure 2 shows the architectural overview of the system. The inner dynamics of each layer were investigated and experimented with using this LA-based big-data application. This architecture takes data via its data bus, runs speed and batch-layer algorithms, then merges the findings from these two layers and updates the dashboard events for display. The dashboard shows the location and velocity of the air vehicle (heatmap), the registered airline company information (word cloud), the nation of the air vehicle (word cloud), the radar chart, the number of ADS-B messages received (plain text), and the number of ADS-B messages processed (plain text). Figure A1, Figure A2, Figure A3 and Figure A4 show all these visualizations.

Eight remote SDRs are placed across Ankara and are utilized as ADS-B signal decoders. Software on the single-board computer decodes the incoming signals and converts the binary data to text-based messages before sending them to the data bus. Apache Kafka is used as the major data bus in this LA, and it is connected to SDR through TCP connections. Spark Streaming + Kafka Integration receiver-less technique is used to connect the data bus to the speed layer. Over Apache Kafka, raw data from several SDRs are combined in D-Streams, followed by ETL and filtering activities at the speed layer. HDFS is also used to save the raw data from Kafka.

In the batch layer, Spark is used to perform ETL and filtering operations, and the results are written again in the HDFS. Following this, Druid’s batch data ingestion process is begun using Druid’s indexing services. Both the Hadoop indexer task and Druid’s indexing service are used to import offline data at the batch layer. The Druid indexing service is utilized if the data are smaller than a few gigabytes; otherwise, a coordination agent starts the Hadoop indexer task for efficient data ingestion to Druid as a serving layer. After ETL processes, the coordination agent is primarily in charge of monitoring and activating batch ingestion activities based on the size of the data offline. This agent starts an indexing process concerning offline data size when a specific amount of offline data reaches a preset value, and the ETL operation is performed. New segments are created regularly, and query-focused views are created at the serving layer.

Some real-time views are superseded by batch views with recent data following batch ingestion. Data delays and repeated data reception conditions are common occurrences. Druid can utilize a data retention policy to delete delayed data after a specific amount of time has passed, and missing data can be rectified via batch-layer ingestion. At the speed layer, a Redis key/value store is linked to the Spark Streaming framework. For updating the aircraft’s information, a publish-subscribe process is used. For aircraft visualization, this database is utilized to buffer data that are more current for 5 min to newly connected clients. Finally, a Node.js service connects to Redis and uses WebSocket technology to transmit all the data to clients. Another web service may be used to query and view historical data. Hadoop HDFS is used on a cluster for fault tolerance and high availability. For the Druid, it also serves as deep storage. On a cluster, YARN and ZooKeeper are used for resource management and coordination, respectively.

### 3.2. Performance Measurement System

In this section, we explain the parts of the performance measurement system. Because big-data systems process huge volumes of data and have a distributed computation architecture, classical debugging and performance testing tools and software cannot be capable of this operation. As a result, we developed a performance measurement system that can execute performance tests. It should be noted that by using the term *performance*, we do not mean metrics related to CPU usage or memory consumption-related metrics; instead, we aim to measure and test the accuracy of the calculated analytics results of the LA system. In the following subsections, we briefly present all parts of the developed testing system.

#### 3.2.1. Data Generation Agent (DGA)

When the data generation software is run, it reads the data production parameters from the configuration file. With these parameters, data repetition, data delay, data transmission rate adjustment, or regular operation are provided. After the software determines the operating mode, it first reads the data from each file through the file reader subunit and writes it to the data reading buffer. When the data reading buffer reaches a certain level, it starts to write the data here to the send buffer according to the mode of operation it is set. Finally, data are read from the sender buffer and sent to Kafka Producer and Kafka Broker at the data transmission rate specified in the configuration file (See Figure 3).

If a data delay is to be made/tested, the data coming from the read buffer is not directly written to the sender buffer. The date and time information of the data to be delayed is taken from the configuration file, and these data are filtered and written to the data reading buffer. Again, a parameter in the configuration file is written to the sender buffer at the specified time. The remaining data, other than filtered data, are written directly to the sender buffer.

When a repetitive operation is requested, the data read from the data reading layer and in the date and time interval specified in the configuration file are written to the repeat buffer. However, other data are transmitted to the transmission buffer as in the typical case. The data in the repeat buffer is regularly written in the transmission buffer at the amount specified in the configuration file and at the message rate specified in the same file and sent to Kafka Broker. With this structure, it is possible to delay and repeat the data at different times. The data generator runs on a single computer with solid-state disks (SSD).

#### 3.2.2. Control and Coordination Agent (CCA)

The control and coordination agent is one of the most critical parts of the LA. It monitors the events taking place in the system, memory, and CPU loads of batch and speed during data processing (operation details given in Section 2.5). It ensures that the batch layer works when necessary and controls data import operations. It also manages the creation of batch and speed views if new analyses are made on immutable data sets, one of LA’s essential features, and new analysis is required due to the change in old analysis results in the time interval. With the start of the system, according to the values obtained by measuring the flow rates of the data coming from the data bus, within the framework of the maximum data-processing capabilities of the system, the BL should be operated without excessive load on the SL. In this way, the amount of data processed by BL is removed from the SL sources so as not to put pressure on the SL’s resources and affect reliability. The specific analysis results are transmitted to the deep storage area. This process is carried out depending on the speed of the data flow coming from the bus, the resource consumption on the SL, and the BL’s data-processing capability. However, this decision, which should be made at the design time, should be chosen according to the application field. To start the operation of the BL, the method of periodically running it can be selected, as well as the approach of running it with the necessary frequency by considering the resource consumption. Although it is possible to use both approaches, resource planning must be considered in a big-data system. The best use of cluster hardware resources is of paramount importance. In this study, instead of periodically processing the data, starting the BL has been adopted by considering the clustering resources.

The coordinated operation of the system is shown in Figure 4. To explain the working principle, it is assumed that the flow rate of the data coming from the data bus does not change during the whole analysis. The operation of the system starting from a state where there is no data on SL and BL (cold start) can be summarized as follows:

There are no data on the system at time t = t${}_{1}$. At this moment, CCA activates the data bus, and SL, and real-time data processing begins. In the time interval t = [t${}_{1}$, t${}_{2}$], only the data in SL are processed and sent to the serving layer. With the accumulation of incoming data during this period, there will be some resource consumption in SL. In the time interval t = [t${}_{2}$, t${}_{3}$], when the consumption of resources in SL reaches a certain level, CCA executes BL. As soon as BL receives the operating command, it performs its analysis of the data that came from the data bus and turned into a stationary state as quickly as possible. In the time interval [t${}_{2}$, t${}_{3}$], all data received in the previous time interval [t${}_{1}$, t${}_{2}$] are processed. When there are no data to be processed, BL stops working and transmits all the information it processes to the serving layer. The area under the flow rate-time graph gives the amount of data processed in a given time interval. When examined in this context, the amount of data that SL processes can be calculated as A${}_{1}$, and the amount of data to be processed by BL can be calculated as A${}_{2}$. To ensure that BL does not fall behind in data processing, it must have A${}_{1}$ data processing in the [t${}_{2}$, t${}_{3}$] time interval, i.e., A${}_{1}$ = A${}_{2}$. However, when A${}_{1\;}$> A${}_{2}$, BL is left behind due to the inability to process SL data in sufficient time. In this case, the load that BL must handle increases with the new data. This situation is repeated in the next period. Finally, BL is far behind SL in processing the data. In this case, LA’s reliable data processing and BL’s correction of errors and deficiencies that may occur in SL are delayed. This situation should never be allowed to occur. The algorithm or the most straightforward design to work in CAA must provide the inequality A${}_{1}$ < A${}_{2}$ inequality.

Even while BL data are processed in the time interval t = [t${}_{2}$, t${}_{3}$], new data continue to arrive SL and BL over the bus. The time interval t = [t${}_{2}$, t${}_{4}$] is the time interval in which the system’s resource consumption is the highest. The most significant resource pressure on clustering occurs in this time interval; while SL is processing data, at the same time, BL starts to process data, so the cluster’s resources are consumed at the same time as SL. Data in this time interval coming from the data bus and processed in SL can only be processed by BL in the time interval t = [t${}_{4}$, t${}_{5}$]. During this period, SL and SK must process all the data received in real time and make them available. The amount of data received during this period can be calculated as the area under the flow-time graph. Therefore, when A${}_{3}$ = A${}_{4}$, data can be appropriately interrogated at the serving layer. If a situation like A${}_{3}$ > A${}_{4}$ occurs this time, it will still fall behind SL in processing BL data. With the other data coming over time, this layer will remain behind, and it will be very late to eliminate errors due to the problems that may occur in SL. This situation is undesirable. Therefore, the amount of data to be processed in BL must meet the condition A${}_{3}$ ≤ A${}_{4}$ to work safely. Safe working will be possible only when A${}_{3}$ ≤ A${}_{4}$. Due to the reasons explained, the CCA must also follow the flow rate of the data coming from the data bus. The correlation between the incoming data rate and the amount of data that the SL can process the reduction of system resources in the SL section of the cluster and the BL’s start time is controlled.

In the case of errors that may occur during the system’s operation, when necessary, the data processing speed of the SL is reduced by reducing the speed of the bus adaptively. Meanwhile, the problematic BL thread is restarted. If the problem is corrected, the bus speed is returned to normal, providing a regular data flow rate. Otherwise, if it fails three times in a row, LA’s data processing is completely stopped. In this case, the bus continues to store data. After the system’s problem is resolved, LA is restarted so those data will be processed again from the last time interval using CCA. This capability is one of the most useful features of LA. Since the data are stored in their raw form, it is possible to process the data by returning to a previous time interval when necessary. In this way, there is no data loss in the system, and eventually, it becomes reliable (Eventually Accurate). The only problem with this situation is the increased need for data storage. However, presently, as disks are very cheap, it is considered a good situation in terms of reliability and data loss.

#### 3.2.3. System Monitoring Agent (SMA)

Each layer and application software in LA transmits the necessary metrics to the Apache Flume Collector service using the Log4J2 software (version 2.13.1) framework. When transmitting data from the Apache Flume Collector service, ETL operations are applied to event logs and metrics in the Interceptor service. All these data are then written to Hadoop HDFS on the one hand and transmitted to be indexed in the Elastic Search. Elastic search accumulates incoming data by indexing them. In these data, visualization operations with Kibana are implemented, and the system is monitored live. In addition to the SL, BL, and data bus event logs, they also send information such as the amount of data processed and the system resource used. On the one hand, this information is transmitted to the CCA to support the decision to initialize the BL. The system monitoring infrastructure is especially used to detect errors and problems that may occur while testing LA with the test and evaluation agent. The errors in the system are eliminated, and the re-test process is started. It is possible to compare the results obtained under the same conditions by correcting and repeating the errors that occur during the tests.

#### 3.2.4. Test and Evaluation Agent (TEA)

As mentioned before, the test and evaluation agent is software in which the serving layer in LA is queried using the data calculated with the MapReduce method in Hadoop. SQL-based queries were created with Apache Hive technology to verify the data in the serving layer. These queries are automatically converted into MapReduce workpieces to generate results. A similar SQL query logic used in SL and BL is used. One-minute time windows were used to generate the results. The resulting results were then imported into the Apache HBase database.

One of the essential features of LA is that the data are kept in raw form. Fixed data are stored in HDFS in raw form. BL processes these static data in HDFS at certain times. The verification data are obtained by processing the same data that have been stopped by the test and verification system. This software is developed in Python language and queries and compares the serving layer and test data for specific time windows. Delays have been avoided due to the development of the software in question by parallelizing it to be multi-thread. More than one thread performs testing and verification at the same time. It takes approximately 10 min to verify the entire serving layer at once.

Analytical metrics are calculated at the one-minute resolution for the test set, as mentioned earlier. The LA system also works on the same analytical metrics in real time and makes the results ready at the serving layer. The test system compares these data with the data produced using a different technology (Hadoop-Hive). In this way, the Recall and Precision values are calculated and saved to another file for visualization. The test and evaluation agent immediately starts to query the serving layer with concise time intervals after the LA starts up and continues the querying process for the next time window when it reaches the previously calculated first Recall and Precision value. This process continues sequentially by shifting to the next time window. Continuous and short-term inquiries are not made in the next time window; only the period between the previous time window and the completed time window is interrogated. This process is that, while the system is being tested, it starts with zero data each time (cold start). Therefore, it should be ensured that the test and evaluation agent system and the LA presentation layer are synchronized.

The system was tested under several data-ingestion conditions with data obtained from SDRs by engaging a DGA. These working conditions were selected to simulate real-world conditions as realistically as possible, summarized as a data rate increase, data rate decrease, instant data rate increase, instant data rate decrease, recurrent data-receiving, and delayed data-receiving. To measure the system performance, latency and the number of correct query results in a time window are calculated. Latency is defined as the time difference between the first ingestion time of data in the system and the visualization time of the same data on the dashboard. According to the performance requirements, this value must be less than 2 s. Predefined analytics queries were prepared and executed in control time windows (3 min), and the results of the queries were compared with the ground truth values. This test was applied sequentially. For the next time window, all experiments were run again. For every test case given above, experiments were executed 50 times to obtain more accurate and statistically significant results.

In Figure 5, the LA operation parameters are shown using conceptual visualization. Here f${}_{D}$ stands for batch layer’s data processing speed, f${}_{A}$ speed layer’s data processing speed, t${}_{1}$, t${}_{2}$, t${}_{3}$, t${}_{4}$, and t${}_{5}$ representing significant system operation times. [t${}_{1}$, t${}_{2}$] is the time interval when only SL is active and working, [t${}_{2}$, t${}_{3}$] is the time interval when both SL and BL are active and working simultaneously, [t${}_{3}$, t_4_] is the time interval when only SL is active and running, and [t${}_{4}$, t${}_{5}$] again another time interval that BL and SL together work and are active. The f${}_{A}$ and f${}_{D}$ are shown here are parameters that can be determined experimentally based on the system’s operational capability. In experiments for the designed system, it was determined that the maximum value of the f${}_{A}$ was 56,000 data/s on average, and the f${}_{D}$ was 130,000 data/s. In a typical operation, the average f${}_{A}$ is set to 25,000 data/s, and the f${}_{D}$ is 54,000 data/s. In addition to these values, the time intervals [t${}_{2}$, t${}_{3}$] and [t${}_{4}$, t${}_{5}$] are set in CCA to be 342 s, and the time interval [t${}_{1}$, t${}_{2}$] and [t${}_{2}$, t${}_{4}$] is 700 s. As can be seen from these parameters, A${}_{1}$, A${}_{2}$, A${}_{3}$, and A${}_{4}$ are calculated to ensure that A${}_{1}$ < A${}_{2}$ and A${}_{3}$ < A${}_{4}$ so that the system has safe data processing and data-processing time can be a maximum of 6 min is determined to be. When about 12,000,000 data are accumulated in SL, it is foreseen that BL will start running. Mathematical modeling of this situation does not give very accurate results. Cluster operation and resource utilization cannot be modeled linearly.

The live data collected from each SDR between 31 August 2015 and 1 October 2015 were recorded in three different files without any processing. Each file in which the said data are written constitutes a data set of approximately 266 GB. Since there may be data that do not arrive on time or may be repeated in the collected data, Hadoop and a MapReduce-based algorithm have arranged the data according to the signal arrival time, and the record number information has been calculated in order not to affect the test results. Although the test results were checked using the required values, delayed and repetitive data that affected the test were prevented in this way. The system’s behavior can be reliably observed by delaying the test data in a controlled manner or sending repeated data on the bus, as will be explained in the following sections. To obtain the data set to be tested, the data between the dates 1–30 September 2015 were filtered with a MapReduce-based algorithm and used to calculate performance criteria in the tests.

The proposed LA-based system is deployed in a cluster. For cluster hardware, 12 Dell Power Edge R730 and 13 Dell Power Edge R320 servers are used. With this cluster, over 6 TB of RAM, more than 500 cores, and more than 0.6 PB of disk space are available for data processing and storage.

The LA developed was first tested to determine the maximum data-processing capacity, and its performance and reliability were examined. Then, the behavior of LA in normal operating conditions was investigated. In addition to this, LA’s behavior and performance in the cases of repetitive data arrival and delayed data arrival, which are the most common problems in real life, were also investigated in detail. The tests in question were carried out using the TEA and the hardware cluster. The results were obtained at least three times for each test, and their consistency was examined. Performance criteria examined in the tests performed were selected as Recall and Precision values. During the tests, the SMA was used to ensure that the same conditions were observed while examining each situation. Thanks to this infrastructure, errors in hardware and clustering were examined, and if there were errors, the test was repeated.

## 4. Results

In this section, we present the results of our experiments for different LA working conditions. In the first part of the experiment, we investigated whether the required LA design specifications were met. In the second of the experiments, we tested and evaluated the accuracy and performance of the LA under six different conditions, such as nominal working condition, repetitive data-ingestion condition, delayed data ingestion case, data ingestion with non-operational batch layer, delayed data ingestion with non-operational batch layer, and finally repetitive data ingestion with non-operational batch layer.

### 4.1. Latency Investigation and Normal Working Condition

Under normal ingestion conditions, an average of 100,000 messages per second was selected to model real air traffic data. The cluster’s computation capability is far beyond these ratings, but air traffic over Ankara does not generate too many ADS-B messages, according to daily observations and statistics. For each data ingestion test, this nominal data rate increased or decreased according to a particular pattern that can accurately model real air traffic. The test and monitoring agent sent test data to the data bus and collected the necessary results and statistics for each test case. All results are given in Table 3 for the first and second time windows.

In Table 3, it can be seen that latency performance criteria are met for all test cases in the first time window. This value is never greater than 401.8 ms for all test conditions. This result shows that the system can effectively process all the incoming data and meets the performance criteria without delay. For this test time window, the number of correct queries was equal to the ground truth. Under delayed data conditions, due to missing data problems, the system query results were not accurately correct. Only 98,790 of 100,000 queries were correctly returned in this time window. These query errors must be corrected in the next time window by the batch layer. Under the recurrent data-receiving conditions, the system had an excellent capability to handle this case effectively. The results show that the LA system works adequately and as expected in all test cases in the first time window.

In the second time window, the latency performance criteria were also met during the tests, and the latency value was never greater than 399.8 ms for all test conditions. This result shows that the system can effectively process all the incoming data and meet the performance criteria without any delay. Results are listed in Table 4. For the second test time window, the number of correct queries was equal to the ground truth values. Under delayed data conditions, the missing data were eventually corrected by the batch layer. Under repetitive data-receiving conditions, the system had an excellent ability to effectively handle this problematic case. The results show that the system works correctly and is eventually prone to data rate changes and problematic data-receiving cases.

Offline data can be queried using the same application. However, in the case of repetitive and delayed data reception, queries are eventually accurate. After the batch-layer operation is completed, query-focused views are overridden by the resulting batch views, and this layer is vital to obtaining accuracy and precision for query results. The data-processing rate under full data load conditions is shown in Figure 6 (also in Figure 7a). Batch and speed layers are seen to work as expected in a complementary manner.

### 4.2. Repetitive Data Ingestion

In the repetitive data retrieval test (in Figure 7b), the LA study was examined in the nominal starting condition. The data received from the data bus are processed in both layers and transmitted to the presentation layer. In this test, the data between 10:00 on 15 September 2015, and 16:00 on 16 September 2015, are sent over the bus repeatedly when BL starts working for the third time. It has been determined that the LA takes approximately 5930 s to process all data. During this period, it was observed that the BL data-processing speed was 57,124 item/s on average, and the data-processing speed of SL was 26,928 item/s. It has been determined that BL’s minimum working time is 353 s, and the maximum working time is 361 s. During the test, it was observed that BL did not lag behind SL in data processing; that is, it did not delay. When performing this test, the system monitoring infrastructure determined that no errors occurred in LA. The test was repeated three times, and the results were determined to be consistent.

### 4.3. Delayed Data Ingestion

In the delayed data case test (in Figure 7c), the LA has been examined in the nominal starting state in which it should work. The data received from the data bus are processed in both layers and transmitted to the presentation layer. In this test, data between 15 September 2015 at 10:00 and 16 September 2015 at 14:00 are delayed when BL starts working for the third time and is sent over the data bus to be forwarded later. It has been determined that LA takes approximately 5920 s to process all data. During this period, it was observed that BL’s data-processing speed was 55,214 items/s on average, and the data-processing speed of SL was 27,111 items/s. It has been determined that BL’s minimum working time is 350 s, and the maximum working time is 365 s. During the test, it was observed that BL did not lag behind SL in data processing; that is, it did not delay. When performing this test, the system monitoring infrastructure determined that no errors occurred in LA. The test was repeated three times to verify the consistency of the results.

### 4.4. Data Ingestion When Batch Layer Is Not Working

The LA’s behavior in the case that the batch data layer was not working (in Figure 7d) was examined under nominal starting operating conditions. The data received from the data bus are processed in both layers and transmitted to the serving layer. In this test, the reaction of LA was examined by preventing BL from running again after it was started only once. In this case, the SL’s data will be processed and made available to the serving layer. The success and reliability of the system in this situation depend only on the success of the SL. It was observed that processing all data with only the SL takes approximately 5943 s. During this period, it was seen that BL’s data-processing speed was zero, and SL’s data-processing speed was 28,558 items/s on average. It has been discovered that the total working time of the BL is 345 s when it is started once. During the performance of this test, the system monitoring infrastructure determined that no errors occurred in the speed layer. The test was repeated five times in a row to verify the consistency of the results. All analysis was carried out at one-minute time window intervals. It has been discovered that the system gives very reliable results even when the BL is not working.

### 4.5. Delayed Data Ingestion When Batch Layer Is Not Working

Finally, In the non-working batch layer and delayed data (latency) test (in Figure 7e), LA was examined in the nominal starting conditions where it should work. Data from the data bus is processed only in the speed layer and transmitted to the serving layer. In this test, the data from 15 September 2015 at 10:00 to 16 September 2015 at 14:00 will be delayed at a time after the BL worked and completed the processing of data, and then transmitted is sent from the bus in such a way. It was determined that the LA processed all the data that lasted approximately 5991 s. During this time, SL’s data-processing speed is seen to be, on average, 27,854 items/s. The one-time operation time of the BL was approximately 345 s. When performing this test, LA was detected with the system monitoring engine, where there was no error. The test was repeated five times, and the consistency of the results was checked.

### 4.6. Repetitive Data Ingestion When Batch Layer Is Not Working

In another experiment, LA’s behavior has been examined (in Figure 7f) in the nominal starting case if the batch layer is not working and then repeated data arrive. Data from the data bus were processed in both layers and transmitted to the serving layer. In this test, the BL was run only once, and the behavior of the results was examined after repeated data were sent to the system. In this case, the data that come to the SL are processed and transmitted to the serving layer. The accuracy and reliability of the system in this situation depend solely on the success of the SL. In this test, data from 15 September 2015 to 10:00 16 September 2015 at 14:00 are sent repeatedly from the data bus after the BL has been run solely once. The LA was determined to process all data with only the SL and lasted approximately 5968 s. During this time, the BL’s data-processing speed was zero, and the SL’s data-processing speed was, on average, 29,225 item/s. Once the BL is run, the total operating time is 345 s. When performing this test, the system monitoring agent quickly detected that no errors occurred. The test was repeated 50 times this time, and the consistency of the results was checked. As already mentioned, the processing of data in the SL and the BL is carried out by the same method. In other words, the code was developed once, and this code was used in both layers in the same way. There is no difference between data coming from multiple sources and data that are repeated multiple times in data processing. It has already been noted that data from several different sources come with a time difference, and the process of converting these data into a single data state is carried out with the algorithm developed.

With these experiments, LA was tested end-to-end using the data produced by the test system. As a result of the experiments, it has been seen that the system works as desired in accordance with the design criteria and fully meets the performance criteria. In addition, it is observed that six different situations that can be encountered in real life can be simulated realistically with the test system.

### 4.7. Lambda Architecture Accuracy Measurement Experiments

In this experiment, the data obtained in real time from each SDR during the period from 31 August 2015, to 1 October 2015, were stored in three distinct files without undergoing any form of analysis or manipulation. Each file containing the aforementioned data represents a data set with an approximate size of 266 gigabytes. To address potential issues such as delayed or duplicated data, Hadoop and a MapReduce-based algorithm have been used to organize the data based on the time of signal arrival. Additionally, the algorithm has calculated the record number information to ensure that it does not impact the precision of the test results. By adhering to the prescribed values, the results of the experiment were verified, thus mitigating the impact of delayed and repetitious data on the test outputs. Reliable observation of the system’s behavior can be achieved by implementing controlled delays in the test data or by transmitting repeated data on the data bus. To acquire the dataset for testing purposes, a MapReduce-based algorithm was employed to filter the data within the time frame of 1–30 September 2015. Subsequently, this filtered data set was utilized to calculate performance metrics in the tests conducted. The metrics calculated that need to be generated by the LA as a result of the analysis are presented in Table 5.

In this experiment, we realistically tested LA for six real-world cases and measured its analytics computation capability to provide accurate results. We implemented nominal working, repetitive data ingestion, delayed data ingestion, data ingestion with no batch layer, repetitive data ingestion with no batch layer, and delayed data ingestion with no batch layer cases with TEA and measured the performance of the LA with several repetitions for each case to obtain statistically significant results.

Precision, which is also known as positive predictive value, refers to the proportion of relevant instances that are included in the retrieved instances. Recall, alternatively referred to as sensitivity, denotes the proportion of pertinent instances that were successfully retrieved. When used independently, precision and recall metrics do not offer significant utility. For example, it is feasible to achieve flawless memory retrieval by retrieving each item. Similarly, it is feasible to achieve a high level of precision by opting for a limited quantity of highly probable items.

To measure the analytical accuracy and performance of LA, we modified and used two well-known classification-based metrics such as precision and recall, for a predefined time window, and definitions of these metrics are given in Equations (Equation 10) and (Equation 11).
(10)$$P\left(T_{w}\right)=\frac{TP}{TP+FP}$$
(11)$$R\left(T_{w}\right)=\frac{TP}{TP+FN}$$

In these equations, *TP*, *TF*, *FP*, *FN* are true positive, true negative, false positive, and false negative values in the specified time window, respectively. $P(T_{w})$ and $R(T_{w})$ are the Precision and Recall values in a specified time window $T_{w}$. We developed and implemented an algorithm to calculate Precision and Recall values in a specified time window and deployed it in the TEA agent. Details of the algorithm are given as pseudocode in Algorithm 1.

In this algorithm, for each time step in a specified data-time window ($T_{w}$), several queries are executed to calculate the analytics metrics given in Table 5 then results are kept in a result-set object for further processing. In the second step, precalculated queries are obtained from the data store. These query results from the data store and the result-set acquired from step 1 are merged by the inner join operation using the date–time stamps. After this operation, using consolidated data and date–time information, Precision and Recall values at that time point are calculated. This operation is executed for each time step, and the results are stored for further analysis. At the end of the full data procession operation, full date–time-based Precision and Recall values are obtained. In the ideal case, *P* and *R* values are required to be close to 1.0 or %100. The lower values of *P* and *R* show that there is a problem in the LA, and the queried and computed analytics results are not fully correct. This method is extremely sensitive to this kind of error that might exist and directly gives very good information about the performance of LA. Even if one data point is missed to process in a given time interval, this results in directly lower values of *P* and *R* than 1.0 or %100.

**Algorithm 1** Calculate Performance Metrics $P(T_{w})$ and $R(T_{w})$ in Time Window $T_{w}$**Require:** $PrecomputedData(T_{w})\notin ⌀$, and $T_{w}>0$**Ensure:** $P(T_{w})\neq 0$, and $R(T_{w})\neq 0$1. $\left\{T_{s}\right\}\leftarrow GenerateTimeSteps\left(T_{w}\right)$2.**for all** $T_{s}^{step}$$\in \{T_{s}\}$ **do**3.  $\{Resultset_{ServingLayer}^{T_{s}^{step}}\}\leftarrow$ ExecuteQueryFromServingLayer($T_{s}^{step}$)4.  $\{Resultset_{Precomputed}^{T_{s}^{step}}\}\leftarrow$ ExecuteQueryFromPrecomputedData($T_{s}^{step}$)5.  $\left\{Resultset_{Temp}^{T_{s}^{step}}\right\}\leftarrow \left\{Resultset_{ServingLayer}^{T_{s}^{step}}\right\}⋈\left\{Resultset_{Precomputed}^{T_{s}^{step}}\right\}$6.  $P(T_{s}^{step})\leftarrow$ CalculatePrecision($T_{s}^{step}$, $\left\{Resultset_{Temp}^{T_{s}^{step}}\right\}$)7.  $R(T_{s}^{step})\leftarrow$ CalculateRecall($T_{s}^{step}$, $\left\{Resultset_{Temp}^{T_{s}^{step}}\right\}$)8.
**end for**
9.

$\left\{T_{w}\right\}\leftarrow {⋃}_{T_{s}=0}^{T_{w}}T_{s}$

10.

$\left\{P\left(T_{w}\right)\right\}\leftarrow {⋃}_{T_{s}=0}^{T_{w}}P\left(T_{s}\right)$

11.

$\left\{R\left(T_{w}\right)\right\}\leftarrow {⋃}_{T_{s}=0}^{T_{w}}R\left(T_{s}\right)$

12.**return** {$T_{w}$}, {$P(T_{w})$}, {$R(T_{w})$}

The system proposed for LA implementation is deployed on a cluster. Cluster hardware consists of 12 Dell Power Edge R730 servers and 13 Dell Power Edge R320 servers. This cluster offers a substantial number of resources for data processing and storage, including more than 6 terabytes of RAM, more than 500 cores, and more than 0.6 petabytes of disk space. The results of the experiments are presented in the following sections.

#### 4.7.1. Nominal Working Case

In this case, LA was started from a zero-processing state (cold start), and then nominal data traffic was applied to the data bus. The data-processing speed of this operation is shown in Figure 7a. It can be seen that both BL and SL are actively working on data processing. The variation in Precision and Recall, which are our performance criteria for eight different analytical queries, according to the analysis date (1–30 September 2015), is shown in Figure 8.

Analysis was performed at one-minute time window intervals. In other words, for every one-minute time window, Algorithm 1 was executed after LA started from a cold state. The Precision and recall values were determined to equal 1.0 (%100) and did not change in a total of nearly 43,200 time windows, and from this point on, LA did not make any errors in any of the time windows. It has been determined that the system gives very reliable results under normal operating conditions from start to end.

#### 4.7.2. Repetitive Data-Ingestion Case

In this experiment, the LA is analyzed in the nominal state in which it should work. In other words, this experiment started while LA was working under nominal conditions, not in a cold start. The data received from the data bus are processed at both batch and speed layers, and then the results are transmitted to the serving layer. Figure 7b shows the variation in the data-processing speed with respect to time in the event of repeated data arrival. In this test, the data between 15 September 2015 at 10:00 and 16 September 2015 at 16:00 are sent from the bus in such a way that they are repeated at the moment when the BL starts working for the third time. The repetitive arrival of these data at the moment when the BL starts to work is one of the worst situations to experience. The correction of the repetitive data will only be possible in the next BL operation. For this reason, the LA behavior was tested by examining this abnormal situation.

The variation in Precision and Recall, which are performance criteria, according to the analysis date (1–30 September 2015), is shown in Figure 9a. Analysis was performed at time intervals of one minute. The Precision and Recall values have been determined to not change in a total of nearly 43,200 time windows, and they are equal to 1 for all time windows. It has been determined that the system gives very reliable results in this operation state and does not make any mistakes. As mentioned above, data processing in SL and BL is developed with the same coding framework. In other words, the code is developed once, and this code is used in both layers. There is no difference between data coming from more than one source and repeated data many times in terms of processing. It has been stated before that the process of converting data from several different sources (SDRs) with time differences into a single data point is carried out with the developed processing algorithm. During the development of the analysis method, i.e., the algorithm, this issue was taken into account, and the reduction (deduplication) procedure was applied within the time windows. In this way, no errors were produced in the outputs of both layers.

During the test, it was observed that BL did not lag behind SL in data processing. When performing this test, the system monitoring infrastructure determined that there was no error in LA. The test was repeated three times, and the results were determined to be consistent.

In cases where this issue is not taken into account, it is foreseen that the results to be obtained will increase the recall value but decrease the Precision value. Data transmitted to the serving layer by SL in this way will not ensure that more than normal data are returned as a result of the query. The reason for this is that, as explained in the design sections before, the data are imported to the SL with a certain scheme, and the operations specified in this scheme are carried out. The cardinality estimation and minimum-maximum operations specified in the diagram are applied to the imported data in real time until the time window is completed. Even if the data come repetitively, the singularity estimation will give the same result because of the operations here. Again, the same results will be obtained thanks to the minimum and maximum retrieval operations when the data are repeated. When thousands of the same data come in, the minimum or maximum of these repetitive data will be the same as the minimum and maximum when it comes once. Therefore, schema-based data import was used as a second safety measure in SL, thanks to the planned design. As long as there is no problem in ETL and other operations in the serving layer, the data that come repeatedly does not affect the retrieval and Precision values in SL. Since this situation was taken into consideration beforehand during the design of the data-processing algorithm, no errors occurred in the queries in the serving layer due to the use of the code developed once in both layers. The analysis result obtained by processing the data sent repeatedly between 15 September 2015, at 10:00 and 16 September 2015, at 16:00 is shown more closely (zoomed) in Figure 9b for this date range.

As a result of this experiment, we discovered that when all layers are functional, using the same coding framework for both BL and SL and by introducing properly designed data-processing algorithms, LA is inherently immune to repetitive data-ingestion problems and can handle this type of incoming data-receiving defects even if data have different arrival times from their sources.

#### 4.7.3. Delayed Data-Ingestion Case

In this test, which examines the delayed arrival of data, the LA is examined in the nominal state in which it should work. In other words, this experiment started while LA was working under nominal conditions, not in a cold start state. The data received from the data bus are processed in both layers and transmitted to the presentation layer. Figure 7c shows the variation in the speed of data processing with time in the event of data delay. In this test, the data between 15 September 2015, 10:00 and 16 September 2015, 14:00, are delayed as soon as BL starts operating for the third time and are sent over the data bus to be transmitted later. As soon as BL starts to work, this state is the worst case in which these data will arrive late. BL’s correction of problematic incoming delayed data will only be possible in the next BL run. For this reason, the LA was tested by examining the worst case.

During the examination, it was observed that BL exhibited comparable data-processing capabilities to SL without any noticeable lag. During the execution of the test, the system monitoring infrastructure detected no errors in the LA. The experiment was conducted in triplicate, and the findings were found to exhibit a high degree of consistency.

As mentioned earlier, the system has been tested in the worst possible condition. Figure 10a,b show the time variation of the recovery and Precision values in the case of data delay and demonstrate the variation of the average precision and retrieval values obtained by querying the records between 15 September 2015, 10:00 and 16 September 2015, 14:00, over 1-min time windows at the maximum 20-min time frame. The Recall and Precision values for a certain period of time become zero, and then these values change with the processing of the data. This change can never reach the value of 1.0. This is because, as mentioned earlier, the data rejection policy in the serving layer is set not to process delayed data. As stated in the LA working method, the serving layer does not import these data when it is processed and transmitted to the serving layer since it comes after the time it should arrive. Again, as stated in the working method of LA, even if the data arrive late, it is collected in its raw form for later processing in the BL. With the operation of the BL at a later time, the data accumulated up to that point are processed. As a result of the operation of this layer, the Recall and Precision values again reach 1.0. As can be understood from this, the designed LA worked without any problems, as it should.

As seen in Figure 10b, the Recall and Precision values caused errors on the minimum distance and maximum distance data at most (zoomed part of the left figure). In these data, the Recall and Precision values remain at very low levels. As noted earlier, the most difficult metric to handle in LA is the minimum and maximum distance. These two metrics, which require complex calculations, cause SL to produce more errors when processed in LA and lower the Recall and Precision values in the serving layer. It seems that complex computational processes are quite sensitive to yield errors in such cases.

The experiment yielded findings indicating that the utilization of both BL and SL in coordination, along with the implementation of well-designed data-processing algorithms, results in inherent immunity to the delayed data-ingestion process in the context of LA. This immunity extends to the handling of delayed data-ingestion issues and the ability to manage incoming data with varying arrival times from their respective sources. It has been stated many times in the previous sections that the LA is ultimately a reliable system (*eventually accurate*). It is seen that a problem that occurs in SL continues until the next BL processing at the latest, and the data obtained as a result of inquiries made from the serving layer when the BL data processing is completed error-free.

### 4.8. Data Ingestion without Batch Layer

This experiment investigates the behavior of the LA for the non-working batch layer. The LA is executed in its intended operational state to assess its functionality. To clarify, the experiment commenced when LA was operating under nominal conditions rather than in a cold start state. The data obtained from the data bus undergoes processing in both layers and is subsequently transmitted to the presentation layer. The results depicted in Figure 7d illustrate the relationship between data-processing speed and time in the event of BL blocking (non-working batch layer). In this test, the reaction of LA was analyzed by running the BL only once and then preventing it from running again. In this case, the data received by the SL will be processed and made available in the serving layer. The success and reliability of the system, in this case, depend only on the success of the speed layer. During the test, the system monitoring infrastructure determined that there was no error in the speed layer. The test was repeated five times, this time to verify the consistency of the results. The variation of the Precision and Recall values, which are performance criteria, according to the analysis date (1–30 September 2015), is shown in Figure 11a,b. The analysis was performed at one-minute time window intervals. It has been determined that the Precision and Recall values do not change in a total of 43,200 time windows, and they are equal to 1.0; from this point on, LA does not make any errors in any of the time windows. It has been determined that the system gives very reliable results even when the BL is not working. As can be seen as a result of this test, it is observed that there is no problem in Recall and Precision values as a result of inquiries made in the serving layer since there is no issue encountered in SL, and there is no such thing as delayed or repetitive data arrivals. In the study conducted to determine the maximum performance limits of the LA, it was observed that it works successfully under overload.

Since its design was made according to the situations in which it will operate under a heavier load, there was no problem in the case that the BL did not work. In addition, since BL, SL, and the serving layer work separately from each other, in other words, since each layer works only for the data type to which it is related and is expert, it is considered that the system works reliably for this condition where BL does not work and there is no delay or repetition of the incoming data. SL and DL process the incoming data and transfer it directly to the serving layer. The data feed of the serving layer in this test was completely over SL. For this reason, it has been determined that the LA, whose BL does not function, works flawlessly and reliably throughout the test process since it does not encounter a problem in data communication and data import operations. How the LA will behave due to the problems that may occur in the data flow in the case BL does not operate will be examined in the following sections.

It is worth noting that in the LA architecture, examining a working condition in which SL does not process data but BL works was unnecessary. The main reason for this is that, in the event of such a situation, the processing of incoming data can only be carried out by BL. Therefore, it was not necessary to examine this operating condition separately since it would not be a study in line with the LA concept but a complete batch data analysis. However, the situation where BL is not operating, and SL is functioning can only be described as SL data analysis. In this working condition, again, there is no data processing in line with the LA concept. However, this working condition has still been examined in the following sections, especially within the scope of this research, and the results have been shared so that the corrective effect of BL can be demonstrated in this very extreme case.

### 4.9. Repetitive Data Ingestion without Batch Layer

In this experiment, we investigate the vase behavior of LA in case the batch data layer does not work and repetitive data are received. The data received from the data bus are processed in both layers and transmitted to the serving layer. Figure 7e shows the variation of the data-processing speed over time when BL is not working. In this test, the BL was not run again after being run only once, and the demeanor of the LA was studied after repetitive data were sent to the system. In this case, the data coming to SL is processed, transmitted to the serving layer, and imported. The success and reliability of the system in this situation depend only on the success of SL’s data processing. The data between 15 September 2015, 10:00 and 16 September 2015, 14:00, were sent repeatedly from the data bus after the BL runs only once.

The variation in Precision and Recall values, which are performance criteria, according to the analysis date (1–30 September 2015), is shown in Figure 12a,b. Analysis was performed at one-minute time window intervals. It has been determined that the Precision and Recall values do not change in a total of nearly 43,200 time windows, equal to 1.0, and from this point on, LA did not produce any miscalculated results in any of the time windows.

As stated previously, the method used for data processing in both SL and BL is identical. For the sake of clarity, the code is created once and subsequently used uniformly across both layers. There is no discernible distinction in the processing of data between multiple sources and the repeated occurrence of data. Previous literature has indicated that data obtained from several distinct sources exhibit temporal disparities. To consolidate these data into a unified form, an algorithm has been devised and used. During the development of the analysis method, specifically the algorithm, no errors were identified in the results obtained from the speed data layer. This can be attributed to the implementation of a reduction process within the designated time windows, which played a crucial role in ensuring the accuracy and reliability of the results. Failure to consider this issue may lead to a potential outcome characterized by an escalation in the return value alongside a decline in the Precision value.

The manner in which the serving layer transmits data does not guarantee an increase in the amount of data returned as a query result beyond the normal amount. The rationale behind this approach is that, as previously outlined in the design sections, the data are imported into the SL using a specific scheme, which then triggers the execution of the operations specified within this scheme. The diagram outlines the utilization of cardinality estimation and minimum-maximum operations on the imported data in real time, continuing until the completion of the designated time window. The cardinality estimation remains consistent despite the repetitive nature of the data due to the consistent application of operations in this context. When data are replicated, the minimum and maximum retrieval operations ensure consistent results. When many identical data points are observed, the minimum and maximum values of these repeated data points will be equivalent to the minimum and maximum values observed when the data point occurs only once. Hence, the implemented design incorporated schema-based data import as an additional safeguard in the serving layer. It is hypothesized that repeated influx of data will not affect Recall and Precision metrics in the SL, provided that there are no issues with ETL processes and other transactions.

In the design strategy of the data-processing algorithm, careful consideration was given to the possible occurrence of repetitive data. Additionally, a schema-based data import method was used in the serving layer. As a result of these measures, no errors were encountered in the queries conducted in the serving layer. The analysis result depicted in Figure 12b illustrates the outcome of the processing of the data that was repeatedly transmitted from 15 September 2015, 10:00 to 16 September 2015, 14:00. This particular representation focuses on (zoomed) the aforementioned time frame, specifically when BL is non-operational and repetitious data are received.

### 4.10. Delayed Data Ingestion without Batch Layer

Finally, in this experiment, the behavior of LA was examined in case the BL does not work, and a data delay occurs at the data bus. Data received from the data bus are processed only in the speed layer and transmitted to the serving layer. In Figure 7f, the variation of the data-processing speed with time is shown in case of data delay with non-operational BL is shown. In this experiment, the data between 15 September 2015, 10:00 and 16 September 2015, 14:00, were sent over the data bus, delayed at a time after the BL was run once, and completed the data processing to be transmitted later. During this experiment, the system monitoring infrastructure determined that there was no error in LA. The test was repeated five times to inspect the consistency of the results.

LA has been tested in the worst possible condition. Figure 13a shows the time variation of the recall and the Precision values in case the BL does not work and a data delay occurs. In this figure, the variation of the average Precision and Recall values obtained by querying the records between 15 September 2015, 10:00 and 16 September 2015, 14:00, over 60 min is shown in Figure 13b. As can be seen in the figure, the Recall and Precision values for a certain period of time become zero, and then these values change with the processing of the data by SL. This change can never reach the maximum performance value of 1.0 for both Precision and Recall metrics. This is because, as mentioned earlier, the data rejection policy in the speed layer is set not to process delayed data. As stated in the working method of LA, the serving layer does not import these data when it is processed and transmitted to the serving layer since it comes after the time it should have arrived. Again, even if the data arrive late, it is collected in its raw form for later processing in the BL. However, because BL works once and then does not work at all, the system has never been able to eliminate the error in the speed layer by processing the delayed data. In the queries made for the date range delayed for 60 min, it was observed that the final Precision and Return values after the data processing remained constant until the end of the experiment for the LA.

As can be seen in Figure 13b, the Recall and Precision values have errors on the minimum distance and maximum distance data at most. It was observed that the Recall and Precision values remained at very low levels. As mentioned earlier, the most difficult metric to handle in LA is the minimum and maximum distance. It is predicted that these two metrics, which require complex mathematical operations and calculations, cause BL to yield more errors when processed in LA and decrease the Recall and Precision values in the speed layer. It is known that complex computational operations are highly error-sensitive in such cases.

## 5. Discussion

This study focuses on an innovative investigation of the LA concept, specifically designed for big-data systems within the realm of air/ground surveillance applications. The research conducted in this study stands out due to its comprehensive investigation of the established framework, surpassing the level of analysis observed in current scholarly works. The system has undergone thorough testing using authentic data, encompassing challenging scenarios, therefore offering a comprehensive assessment of its performance across diverse conditions. This innovative methodology has not only enhanced our comprehension of LA but also established a novel benchmark for forthcoming investigations in the domain [47].

The research focuses on the difficulties encountered during the implementation of LA [23], including the need to create separate software or codes for data processing, as well as the operational complexities involved in managing and resolving issues with two systems. The utilization of the technologies and methods outlined in this study has effectively obviated the necessity of creating distinct software or codes [17]. Research has demonstrated the feasibility of utilizing a dual-layer approach in software development, wherein a single set of software/code can be employed for both data processing and algorithm development purposes.

The experiments were meticulously designed to simulate real-world challenges that a big-data system might encounter, and the results have yielded significant insights into the capabilities of LA. Our initial experiment served as a baseline, establishing the efficiency of LA under normal operating conditions. The data processing rate was found to be highly efficient, confirming that LA is adept at handling large volumes of data in real time. This is a critical requirement in aerospace applications, where timely and accurate data processing can be a matter of utmost safety. The architecture demonstrated its capability to process data at a rate that meets the demands of such high-stakes environments [6], thus highlighting its potential as a reliable solution for industries that operate under stringent conditions. One of the notable strengths of LA, as revealed by our experiments, is its robustness against repetitive data-ingestion problems. The architecture, by virtue of using the same coding framework for both batch and speed layers and employing well-designed data-processing algorithms, proved to be inherently immune to issues related to repetitive data ingestion. This is a significant finding, as data duplication is a common issue in big-data systems. Our results show that LA can handle such scenarios gracefully, effectively eliminating redundancies and ensuring the integrity of the processed data.

Our experiments also delved into a worst-case scenario in which data are delayed as soon as the batch layer starts operating. The results were consistent and revealed that the batch layer exhibited comparable data-processing capabilities to the speed layer without any noticeable lag [13]. This is a critical finding, as it demonstrates the resilience of LA in challenging conditions. It affirms the architecture’s suitability for applications where data reliability is paramount, showcasing its ability to maintain high levels of accuracy and consistency even when faced with delayed data ingestion. In scenarios where the batch layer was not operational, our experiments showed that the system remained resilient and capable of handling data processing through the speed layer alone. This is a significant result as it indicates that LA can continue to function effectively even when one of its components is compromised [14]. This highlights the fault-tolerant nature of LA, which is a crucial attribute for systems deployed in environments where continuous operation is essential. Our study also ventured into extreme cases, such as when data are delayed, and the batch layer is not working simultaneously. These experiments were particularly insightful as they allowed us to observe the corrective effect of the batch layer under very challenging conditions. The results of these tests further solidify the reliability and resilience of LA under various operational circumstances.

The experiments carried out have provided evidence of the durability and dependability of the developed ADS-B data analysis system in various circumstances. Under typical ingestion circumstances, the system demonstrated efficient data-processing capabilities and met performance standards successfully without noticeable delays. Even in atypical circumstances, such as instances of delayed data or recurring data-receiving problems, the system demonstrated exceptional proficiency and precision. The findings of this study provide compelling evidence to support the notion that the LA system is inherently reliable and resilient against various challenging operational circumstances [12].

Our experiment results demonstrate that LA is an eventually accurate system, even under several problematic working conditions. These findings align with the claims of previous literature, but for the first time, we provide empirical evidence to support these previous claims. Our empirical findings cover a much broader range of data-ingestion cases, including problematic ones and conditions that have not been investigated up to now.

In summary, our experiments have demonstrated that LA is a highly resilient and reliable system for big-data processing. It can effectively handle a variety of challenging scenarios, making it a promising solution for complex, real-time data-processing tasks, particularly in critical domains such as aerospace. Our study contributes to the field by empirically validating the capabilities of LA, therefore providing a strong foundation for its adoption in various industries that require robust and reliable big-data processing solutions.

The research presented in this study is distinguished by its original interpretation [1] of the LA concept and the comprehensive evaluation of the constructed framework using empirical data, which encompasses scenarios of exceptional nature. To the best of our current understanding, this study represents the inaugural attempt to assess the efficacy of a big-data system utilizing Lambda Architecture implementation in addressing real-world challenges within the aerospace field [8].

## 6. Conclusions

This study introduces an innovative approach to evaluating LA, which may have potential applications in other big-data frameworks. The methodology encompassed the evaluation of prospective technologies at all levels of the layered architecture, examination of design limitations specific to each layer, and the implementation of a uniform software development framework across multiple layers (utilizing identical code for different layers, known as SC-FDL). The proposed LA was implemented in a practical case study involving air/ground surveillance [15,16,18,19]. Subsequently, performance trials were carried out in various scenarios to evaluate its effectiveness using real-world data-intake situations.

The novelty of this research lies in its empirical approach. Our experiment results demonstrate that LA is an eventually accurate system, even under several problematic working conditions. These findings align with the claims of previous literature [12,13,14], but for the first time, we provide empirical evidence to support these previous claims. Our empirical findings cover a much broader range of data-ingestion cases, including problematic ones and conditions that have not been investigated up to now. To the best of our knowledge, this study proposes for the first time a comprehensive LA testing methodology. This methodology has been applied to real-world data-ingestion operations for air/ground surveillance and monitoring, providing a practical demonstration of its effectiveness. This study not only contributes to the existing body of knowledge on LA but also addresses a significant literature gap. By providing a novel methodology for testing LA and demonstrating its effectiveness in real-world applications, this study paves the way for further research in this area. However, a gap in the literature persists, particularly in the absence of a testing and measuring method for the performance and accuracy of LA. This study aims to fill this gap by proposing a novel methodology for testing LA, which could potentially apply to other big-data architectures.

Notwithstanding the notable contributions of this study, certain aspects could be improved. For example, the research could be enhanced by conducting a more comprehensive examination of the influence of varying data-ingestion rates on the efficacy of the LA system. Furthermore, the research could investigate the possibility of incorporating alternative big-data technologies into the LA system to augment its efficiency and functionalities.

In terms of future research, it would be interesting to explore the applicability of the proposed LA system to other domains beyond air/ground surveillance. This would help to further validate the universality of the proposed testing methodology. Moreover, further studies could investigate the potential of integrating machine learning algorithms into the LA system to enhance its data processing and analytical capabilities. Another primary limitation of the current study is the lack of mathematical proof or evidence for SC-FDL. Although we have used this approach, future studies should aim to develop an algorithmic evaluation methodology that can provide more concrete empirical evidence. Finally, as the field of big data continues to evolve, so will the technologies and methodologies used in its analysis. Future research should therefore continue to explore and evaluate new approaches to big-data architecture, ensuring that the methodologies we develop remain relevant and effective in the face of rapid technological change. The findings of this study provide a solid foundation for future research endeavors.

In summary, the findings of this research offer compelling evidence that substantiates the efficacy and dependability of the LA system developed in the context of air/ground surveillance applications. The research presented in this study is distinguished by its original interpretation of the LA concept, thorough testing conducted under various conditions, and the successful demonstration of the system’s performance and capabilities. These factors collectively highlight the novelty and significance of the research. This study makes a valuable contribution to the field of big-data research while also offering a practical solution to the real-world issue of air/ground surveillance.

## Figures and Tables

**Figure 1 sensors-23-07580-f001:**
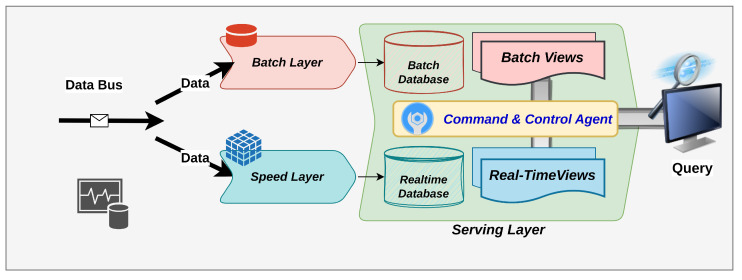
Conceptual diagram of Lambda Architecture.

**Figure 2 sensors-23-07580-f002:**
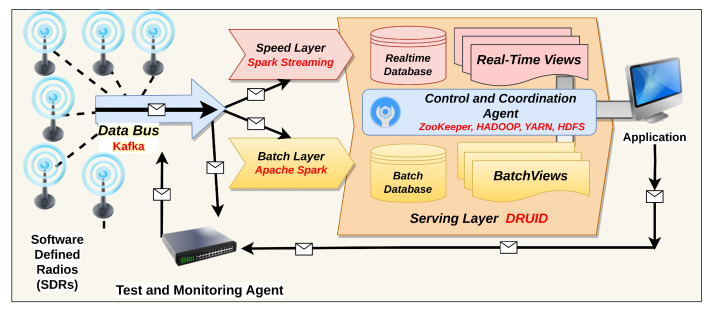
Lambda architecture for air traffic monitoring applications.

**Figure 3 sensors-23-07580-f003:**
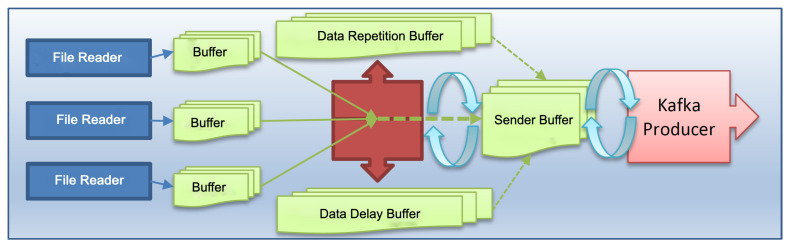
Data generation agent flow diagram.

**Figure 4 sensors-23-07580-f004:**
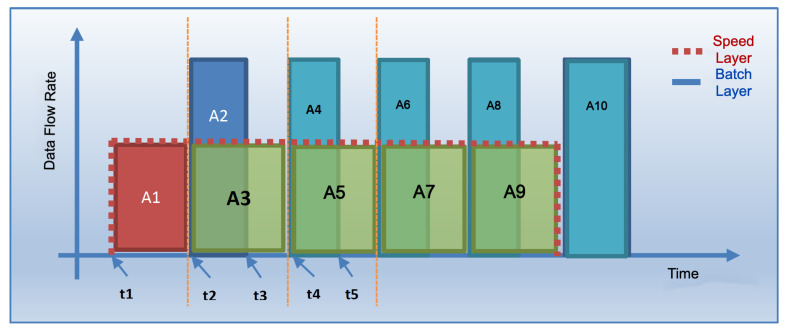
Coordinated operation of BL and SL with CAA.

**Figure 5 sensors-23-07580-f005:**
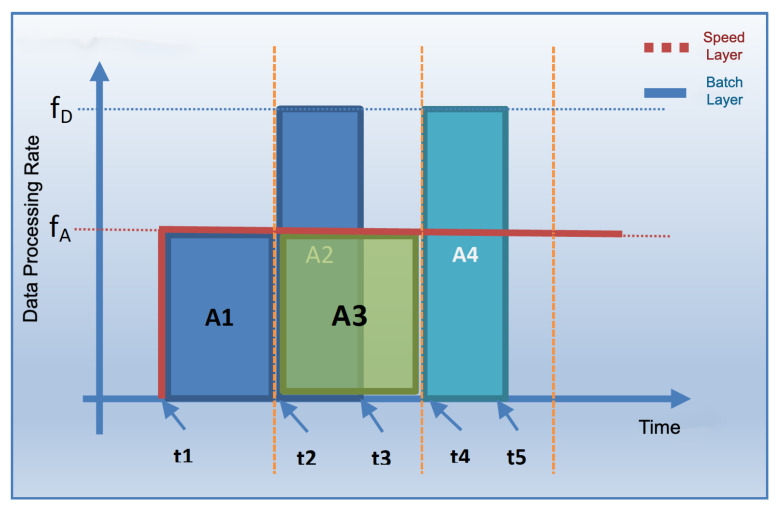
Conceptual visualization of CCA operation parameters.

**Figure 6 sensors-23-07580-f006:**
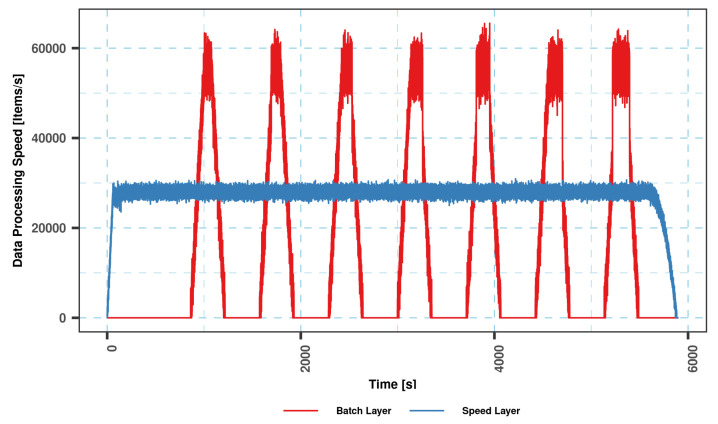
Data processing speed under normal conditions.

**Figure 7 sensors-23-07580-f007:**
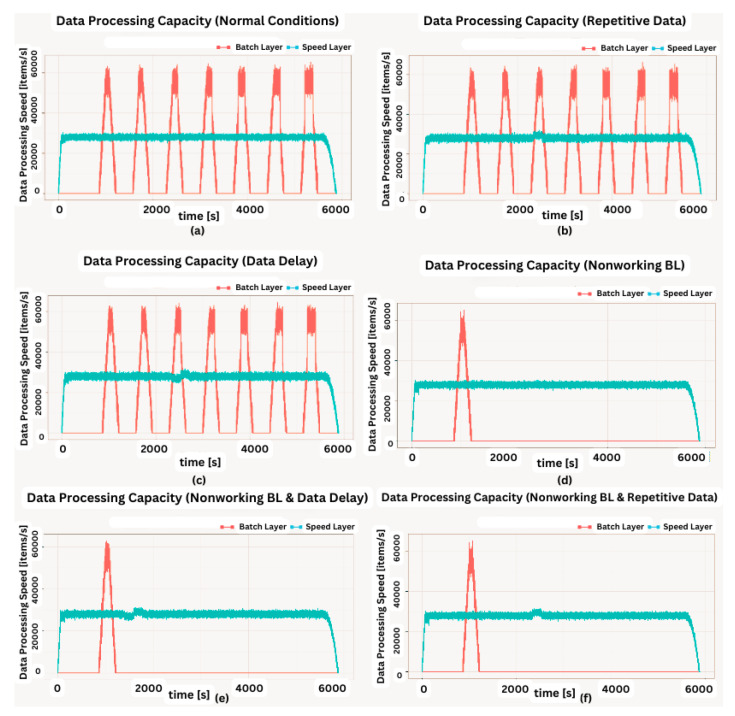
Data processing results. (**a**) Normal conditions; (**b**) Repetitive data ingestion; (**c**) Delayed data ingestion; (**d**) LA with BL not working case; (**e**) LA with delayed data and BL not working case; (**f**) LA with repetitive data and BL not working case.

**Figure 8 sensors-23-07580-f008:**
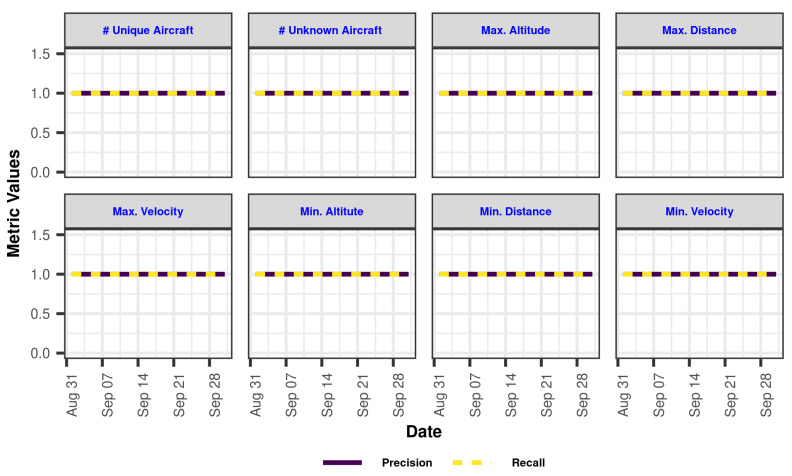
Variation of performance metrics for the nominal working case.

**Figure 9 sensors-23-07580-f009:**
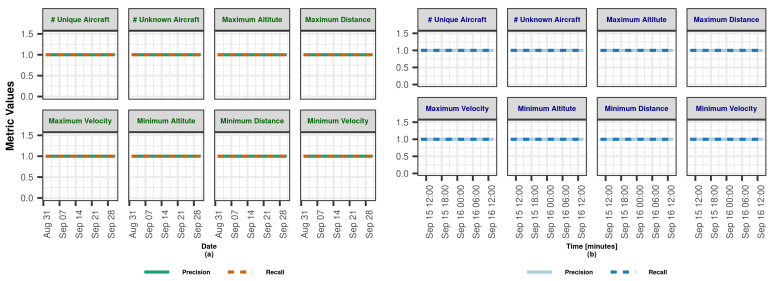
Variation of performance metrics for repetitive data-ingestion case. (**a**) Full View; (**b**) Zoomed View.

**Figure 10 sensors-23-07580-f010:**
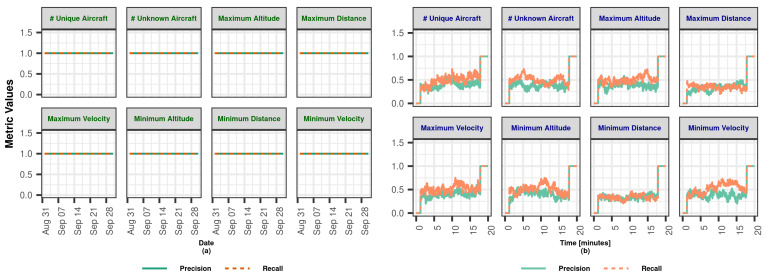
Variation of performance metrics for delayed data-ingestion cases. (**a**) Full View; (**b**) Zoomed View.

**Figure 11 sensors-23-07580-f011:**
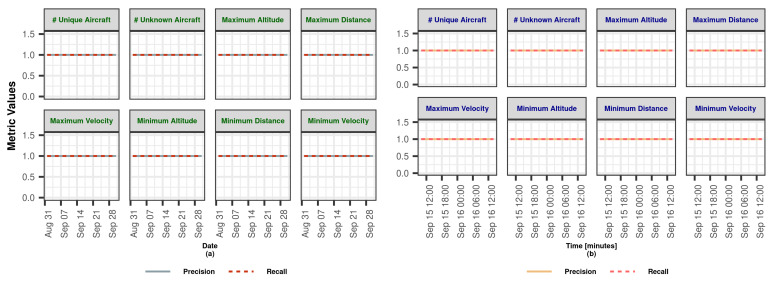
Variation of performance metrics for data ingestion when the batch layer stops working. (**a**) Full View; (**b**) Zoomed View.

**Figure 12 sensors-23-07580-f012:**
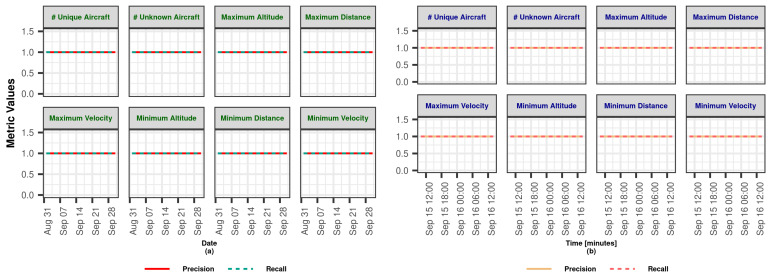
Variation of performance metrics for repetitive data ingestion when the batch layer stops working. (**a**) Full View; (**b**) Zoomed View.

**Figure 13 sensors-23-07580-f013:**
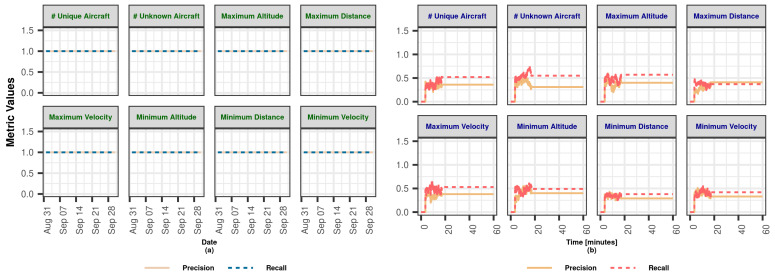
Variation of performance metrics for delayed data ingestion when the batch layer stops working. (**a**) Full View; (**b**) Zoomed View.

**Table 1 sensors-23-07580-t001:** Selected big-data technologies for the application.

Layer/Component Name	Big-Data Technology
Batch Layer	Hadoop/Apache Spark
Speed Layer	Spark Streaming
Serving Layer	Druid
Data Bus	Apache Kafka
Layer Coordination and Control Agent	YARN Application
Monitoring Agent	YARN

**Table 2 sensors-23-07580-t002:** Comparison of The Proposed and Traditional LAs in Respect to Big-Data Technologies.

Items	Traditional Lambda Architecture	Proposed Lambda Architecture
Batch Layer	Storage-HDFS	Storage-Kafka
	Compute-MapReduce	Compute-Apache Spark
Speed Layer	Apache Storm	Spark Streaming
Serving Layer	ElephantDB, Apache Cassandra	Apache Druid
Data Bus	Apache Kafka	Apache Kafka
Code Reusability	Low	High
Complexity	High	Low
Maintenance	High	Low
Learning Curve	High	Low
Cost	High	Low

**Table 3 sensors-23-07580-t003:** Test results for different data-ingestion conditions for 1st time window.

Test Case	Average Latency	Query Results	
	**(ms)**	**Measured**	**Ground Truth**
Normal Conditions	322.5 ± 45	100,000	100,000
Data Rate Increase	401.8 ± 66	100,000	100,000
Data Rate Decrease	323.6 ± 23	100,000	100,000
Instant Data Rate Increase	369.8 ± 89	100,000	100,000
Instant Date Rate Decrease	348.4 ± 56	100,000	100,000
Delayed Data	326.5 ± 34	98,790	100,000
Repetitive Data	328.7 ± 78	100,000	100,000

**Table 4 sensors-23-07580-t004:** Test results for different data-ingestion conditions for 2nd time window.

Test Case	Average Latency	Query Results	
	**(ms)**	**Measured**	**Ground Truth**
Normal Conditions	328.5 ± 61	100,000	100,000
Data Rate Increase	399.8 ± 67	100,000	100,000
Data Rate Decrease	313.2 ± 45	100,000	100,000
Instant Data Rate Increase	352.8 ± 62	100,000	100,000
Instant Date Rate Decrease	338.4 ± 44	100,000	100,000
Delayed Data	336.5 ± 82	100,000	100,000
Repetitive Data	335.7 ± 35	100,000	100,000

**Table 5 sensors-23-07580-t005:** Definition of analytic metrics used for tests.

Calculated Analytical Metrics	Definition (In a Predefined Time Window)
Avg. Number of Unique Aircraft	Unique aircraft count/total detected aircraft count
Avg. Number of Unknown Aircraft	UNK aircraft count/total detected aircraft count
Min. Aircraft Speed	Min. speed in all detected aircraft
Max. Aircraft Speed	Max. speed in all detected aircraft
Min. Aircraft Distance	Min. haversine distance between SDR and aircraft
Max. Aircraft Distance	Max. haversine distance between SDR and aircraft
Min. Aircraft Altitude	Min. altitude in all detected aircraft
Max. Aircraft Altitude	Max. altitude in all detected aircraft

## Data Availability

Not applicable.

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
