# Peer review of "Performance Analysis of Lambda Architecture-Based Big-Data Systems on Air/Ground Surveillance Application with ADS-B Data"

_sensors, 2023, doi:10.3390/s23177580_

Round 1

Reviewer 1 Report

Abstract can be improved with more clarity.

References are sufficient & up-to-date.

Figures 7 text size is small, can be increased.

Figure A1 is not visible properly, can be improved.

Minor editing of English language required.

Check for some grammatical errors.

Reviewer 2 Report

1.     ALL Abbreviations should be defined in full the first time they appear in the title, abstract, main text, and figure or table captions, even if they are well known in the field.

The first time you use an abbreviation in the text, present both the spelled-out version and the short form.

The syntax is: The fully spelled out name (abbreviation)

Lambda Architecture (LA)

Internet of Things (IoT)

Please check for all other abbreviations in your manuscript.

2.     In the Problem Definition section provide a comparison table of Literature Review to clear your study gap

3.     It is better to compare the results with other similar recent works not compare only with Traditional Lambda Architecture.

4.     Add a separate section for the conclusion and future work

The paper required minor revision 

Reviewer 3 Report

This paper evaluated the performance and accuracy of Lambda Architecture (LA) by proposed a novel methodology, having a potentially applicable to other big data architectures.

1.      In Abstract section, the content focused on the proposed novel methodology to evaluate the performance of LA, whereas the title of the paper focused on the performance analysis, not the proposed evaluate method.

2.       In the section 2.4 Same Coding for Different Layers (SC-FDL) Approach, more issues have been described for the LA, while, what is the detail of SC-FDL approach?    

3.      As the paper described, the mathematical proof or concrete empirical evidence for SC-FDL cannot be provided currently, how to verify the validation of SC-FDL as an evaluation methodology?

      4.Conclusion is needed.

Sentences need to be descibed concisely and clearly
